# Genome-wide effects of the antimicrobial peptide apidaecin on translation termination in bacteria

**Kyle Mangano[1,2†], Tanja Florin[1†], Xinhao Shao[2], Dorota Klepacki[1], Irina Chelysheva[3], Zoya Ignatova[3], Yu Gao[2], Alexander S Mankin[1,2*], Nora Vázquez-Laslop[1,2*]**

[1]Center for Biomolecular Sciences, University of Illinois at Chicago, Chicago, United States; [2]Department of Pharmaceutical Sciences, University of Illinois at Chicago, Chicago, United States; [3]Institute of Biochemistry and Molecular Biology, University of Hamburg, Hamburg, Germany

**Abstract** Biochemical studies suggested that the antimicrobial peptide apidaecin (Api) inhibits protein synthesis by binding in the nascent peptide exit tunnel and trapping the release factor associated with a terminating ribosome. The mode of Api action in bacterial cells had remained unknown. Here genome-wide analysis reveals that in bacteria, Api arrests translating ribosomes at stop codons and causes pronounced queuing of the trailing ribosomes. By sequestering the available release factors, Api promotes pervasive stop codon bypass, leading to the expression of proteins with C-terminal extensions. Api-mediated translation arrest leads to the futile activation of the ribosome rescue systems. Understanding the unique mechanism of Api action in living cells may facilitate the development of new medicines and research tools for genome exploration.

**\*For correspondence:**
shura@uic.edu (ASM);
nvazquez@uic.edu (NVL)

†These authors contributed
equally to this work

**Competing interests:** The authors declare that no competing interests exist.

## Introduction

The ribosome translates mRNA into protein and represents one of the main antibiotic targets in the bacterial cell. Various steps of protein synthesis are inhibited by natural and synthetic antibacterials (*Lin et al., 2018*; *Polikanov et al., 2018*; *Wilson, 2009*). A number of antibiotics impede the initiation of translation by preventing mRNA binding or departure of the ribosome from the start codon (*Lin et al., 2018*; *Polikanov et al., 2018*; *Wilson, 2009*). Numerous inhibitors affect translation elongation by interfering with mRNA decoding, peptide bond formation, translocation, or passage of the nascent protein through the exit tunnel (*Lin et al., 2018*; *Polikanov et al., 2018*; *Vázquez-Laslop and Mankin, 2018*; *Wilson, 2009*). However, no antibiotics were known to specifically target the termination step of protein synthesis. It was only recently that apidaecin, an antimicrobial peptide from honeybees (*Casteels et al., 1989*), was described as the first antibiotic specifically targeting translation termination (*Florin et al., 2017*).

Apidaecin is an 18-amino acid proline-rich antimicrobial peptide (PrAMP). Various PrAMPs are produced by insects and mammals to protect the host from bacterial infections (*Graf and Wilson, 2019*; *Scocchi et al., 2011*). The antimicrobial activity of PrAMPs is based on their ability to bind to the bacterial ribosome and inhibit protein synthesis (*Castle et al., 1999*; *Krizsan et al., 2014*; *Mardirossian et al., 2014*). PrAMPs bind in the vacant nascent peptide exit tunnel of the large ribosomal subunit (*Gagnon et al., 2016*; *Roy et al., 2015*; *Seefeldt et al., 2016*; *Seefeldt et al., 2015*). Most of the PrAMPs described to date invade the peptidyl transferase center (PTC) and by hindering the binding of the first elongator aminoacyl-tRNA, arrest the ribosome at the start codon (*Gagnon et al., 2016*; *Seefeldt et al., 2016*).

The action of apidaecin is principally different. Even though this PrAMP also binds in the exit tunnel and closely approaches the PTC, it does not directly obstruct the catalytic site (*Florin et al., 2017*; *Krizsan et al., 2015*). Biochemical analyses carried out with the synthetic peptide Api137 (*Berthold et al., 2013*; referred to, throughout, as Api) showed that Api traps deacyl-tRNA in the P site and class one release factors (RF1 or RF2) in the A site. The model that emerged from these in vitro studies (*Florin et al., 2017*) postulated that Api can bind only after the ribosome has reached the stop codon, associated with RF1/RF2, and released the fully-synthesized protein, because only then the exit tunnel becomes available for Api binding. This view implied that Api specifically targets the post-release ribosome. Strikingly, however, addition of Api to a cell-free translation system could also lead to the accumulation of unhydrolyzed peptidyl-tRNA, indicating that Api is also able to interfere with peptidyl-tRNA hydrolysis at stop codons (*Florin et al., 2017*). To explain this observation, it was proposed that Api-mediated trapping of RF1/RF2 on the post-release ribosomes would prevent the release of completed proteins at the remaining Api-free ribosomes (*Florin et al., 2017*).

The proposed mechanism of Api action has been inferred primarily based on in vitro biochemical and structural studies involving a limited number of artificial substrates and reporters. Furthermore, there is a seeming discrepancy regarding the in vitro and in vivo modes of Api action, since the translation of a GFP reporter protein was significantly inhibited by Api in bacterial cells, whereas expression of the same reporter in a cell-free system was only marginally affected even at high Api concentrations (*Krizsan et al., 2015*; *Krizsan et al., 2014*). Therefore, the consequences of Api action upon translating ribosomes in the living cell have remained unknown.

To investigate the genome-wide impact of Api on translation in bacterial cells we used ribosome profiling (Ribo-seq). Our analysis revealed that Api globally inhibits translation termination. The general arrest of translation at the end of the ORFs leads to a pronounced queuing of elongating ribosomes behind those occupying stop codons. Strikingly, the treatment of *E. coli* with Api results in a dramatic stop codon readthrough that generates proteins with C-terminal extensions. Our data also show that, trying to mitigate the pervasive Api-induced readthrough, cells engage the ribosome rescue systems, which nevertheless remain ineffective because of the Api action.

## Results

### Api redistributes ribosomes within mRNAs

Prior to carrying out the Ribo-seq experiment, we first optimized the conditions of Api treatment to achieve a significant antibiotic effect while minimizing undesirable secondary events. Incubation of *E. coli* (BL21) cultures for 1 min with increasing concentrations of Api resulted in severe inhibition of protein synthesis (*Figure 1—figure supplement 1A*). Exposure to 1.25 µM of Api, which corresponds to the minimal inhibitory concentration (MIC) that prevents cell growth, reduced translation by as much as 75%. Treatment for 2 min with 4× MIC diminished protein synthesis to nearly 10% of the control and extending the treatment to 10 min achieved ~94% inhibition (*Figure 1—figure supplement 1B*). These results confirmed the previous assertion that Api acts as a general inhibitor of bacterial translation (*Castle et al., 1999*; *Krizsan et al., 2014*) and guided the selection of the treatment conditions for the subsequent experiments.

We then proceeded to the Ribo-seq experiments to obtain an unbiased view of how treatment with Api alters the global distribution of translating ribosomes in cellular mRNAs (*Ingolia et al., 2009*; *Oh et al., 2011*). Exponentially-growing *E. coli* cells were incubated for 5 min with 4× MIC of Api, conditions that led to nearly complete inhibition of translation (*Figure 1—figure supplement 1B*), and ribosome-protected mRNA fragments (ribosome footprints) were isolated, sequenced, and mapped to the *E. coli* genome (*Becker et al., 2013*; *McGlincy and Ingolia, 2017*). Exposure to Api caused a dramatic redistribution of ribosomes along mRNAs. The general pattern observed within individual genes revealed a buildup of ribosome density toward the ends of the ORFs, often spilling into the 3′-intergenic regions (*Figure 1A*, *Source data 1*). In operons with narrowly spaced genes, the ribosome density was primarily observed at the junctions spanning 3′-terminal codons of the upstream ORF and 5′-proximal codons of the downstream ORF (*Figure 1B*). Overall, these effects are drastically different from those observed previously with the antibiotic retapamulin, a specific inhibitor of translation initiation, which generates precise and sharp peaks representing ribosomes occupying start codons (*Meydan et al., 2019*). To mitigate the uncertainty of assigning reads at the

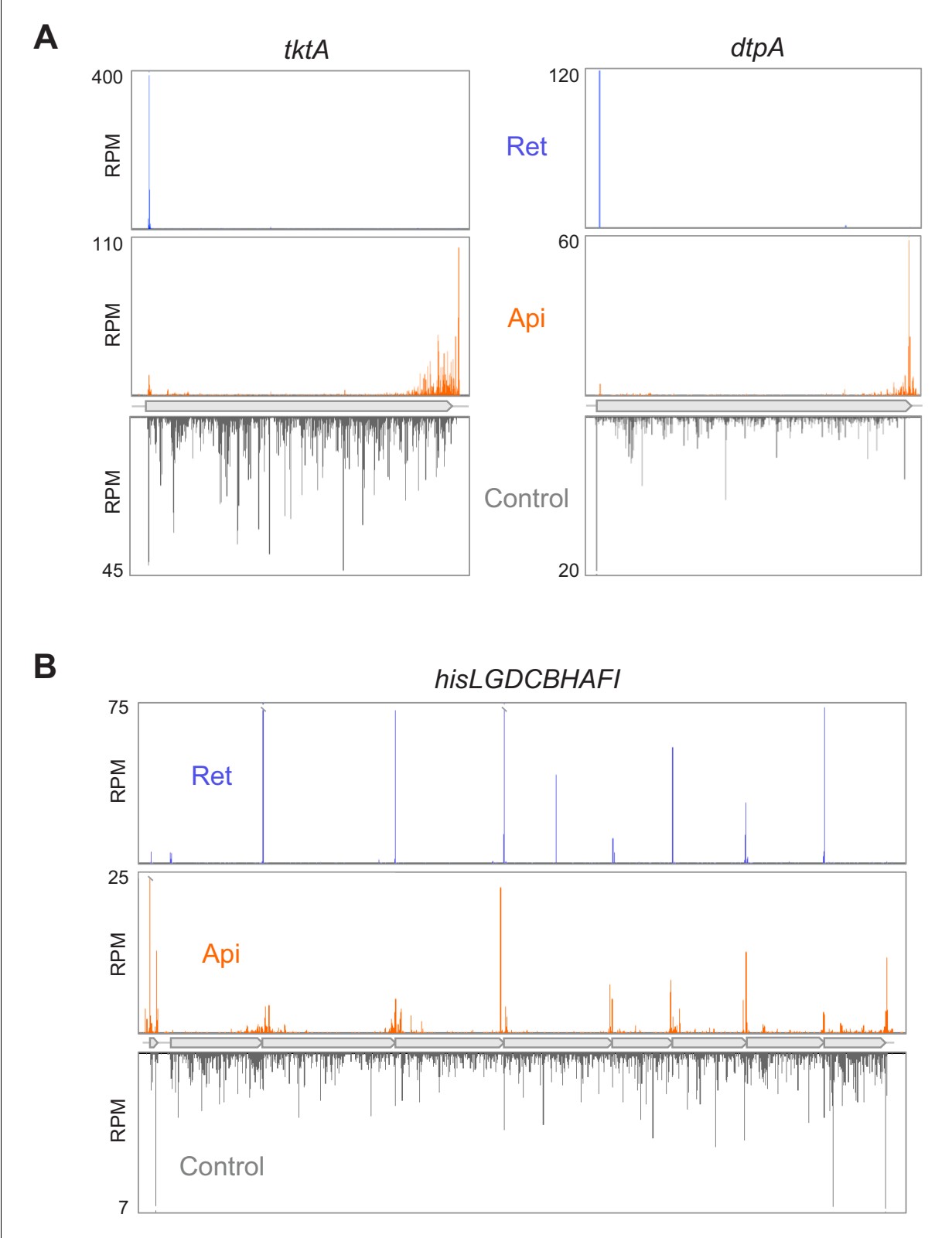

**Figure 1.** Api redistributes ribosomes toward the ends of mRNA ORFs. (**A**) Comparison of ribosome footprint density in the *tktA* and *dptA* ORFs in untreated *E. coli* cells with that in cells treated with either Api or the translation initiation inhibitor retapamulin (Ret). The position of the ribosome footprint was assigned to the first nucleotide of the codon positioned in the P site, presumed to be at a distance of 16 nt from the 3' end of the read

*Figure 1 continued on next page*

*Figure 1 continued*

(*Woolstenhulme et al., 2015*). (B) Ribosome footprint density within genes of the *his* operon in cells treated with no antibiotic, Api or Ret. The Ret Ribo-seq data are from *Meydan et al., 2019*.

The online version of this article includes the following figure supplement(s) for figure 1:

**Figure supplement 1.** Api inhibits global protein synthesis in *E. coli* cells.

junctions of adjacent or overlapping genes in the operons, we limited our subsequent analysis to actively translated genes that are separated by at least 50 nt from the nearest gene (see Materials and methods). The metagene analysis of ribosome coverage near the stop codons of the well-separated genes (*Figure 2A*) revealed several major effects of Api action that will be discussed in the following sections.

## Api acts as a global inhibitor of translation termination

A prominent outcome of Api treatment is a significant (~tenfold) increase of the average normalized ribosome density at stop codons (*Figure 2A*). Approximately 11% of all the ribosome footprints in Api-exposed cells map to stop codons. This result reveals Api as a potent global inhibitor of translation termination in bacterial cells and expands the previous findings of the in vitro experiments which showed that Api could arrest ribosomes at the stop codon of several model genes during cell-free translation (*Florin et al., 2017*).

Given the disparity in ribosome occupancy at and around termination sites in individual genes (*Figure 1A*), we computed stop codon (SC) scores to systematically assess the extent of Api-induced ribosome stalling in these regions. The SC score reports the ribosome density within the last three codons of the genes (i.e., the last two sense codons and the stop codon of the ORF) relative to the average density across the entire gene (*Figure 3A*, *Figure 3—figure supplement 1*). In the Api-treated cells, the SC scores of 97% of genes shifted to higher values compared to the untreated control. Although SC scores varied significantly between the genes, more than half (56%) of the genes in the Api sample exhibited values above 8 ($\log_2 = 3$), whereas only 1% of the genes in the control cells showed such high scores, reflecting the dramatic increase in stop codon occupancy caused by Api (*Figure 3A*).

We considered the possibility that the SC score variation between the genes could be due to a differential action of Api toward RF1 and RF2. Indeed, the Api sample SC scores at the RF2-specific UGA codons were somewhat lower than those at the RF1-specific UAG codons, but the difference was fairly modest and generally followed the trend observed in the untreated control (*Figure 3—figure supplement 2*). This result shows that Api efficiently acts upon both class 1 RFs, thereby contradicting a recent suggestion that Api preferentially traps RF1 over RF2 (*Kuru et al., 2020*). In search for other features that could be associated with a more pronounced Api-induced ribosome stalling at the ends of specific genes, we analyzed the C-terminal sequences of the proteins encoded in ORFs with high SC-scores (*Figure 3B*). The pLogo analysis showed statistically significant prevalence of glycine residues at the C-termini of such proteins, a tendency not observed in high SC-scoring ORFs in the untreated cells. This result indicates that Api does not simply exacerbate inherent difference in translation termination efficiency between the genes, but rather acts in a context-specific manner.

No remarkable sequence context features were detected at the regions downstream of the stop codons of the high-SC score genes (*Figure 3—figure supplement 3A*). Similarly, none of the other examined properties, including mRNA abundance, its propensity to form stable secondary structures, or the number of ribosomes per mRNA molecule show any strong correlation with the SC scores of the genes in the Api-treated cells (*Figure 3—figure supplement 3B*).

Taken together, our data revealed that the primary mode of Api action in the cell is inhibition of translation termination and that this effect is influenced by sequence context.

## Api-induced translation arrest at stop codons results in queuing of the elongating ribosomes

At least three distinct waves of ribosome density are observed in the metagene profile upstream from the stop codons in the Api sample (*Figure 2A*). The crests of the waves are spaced by ~27 nt, a

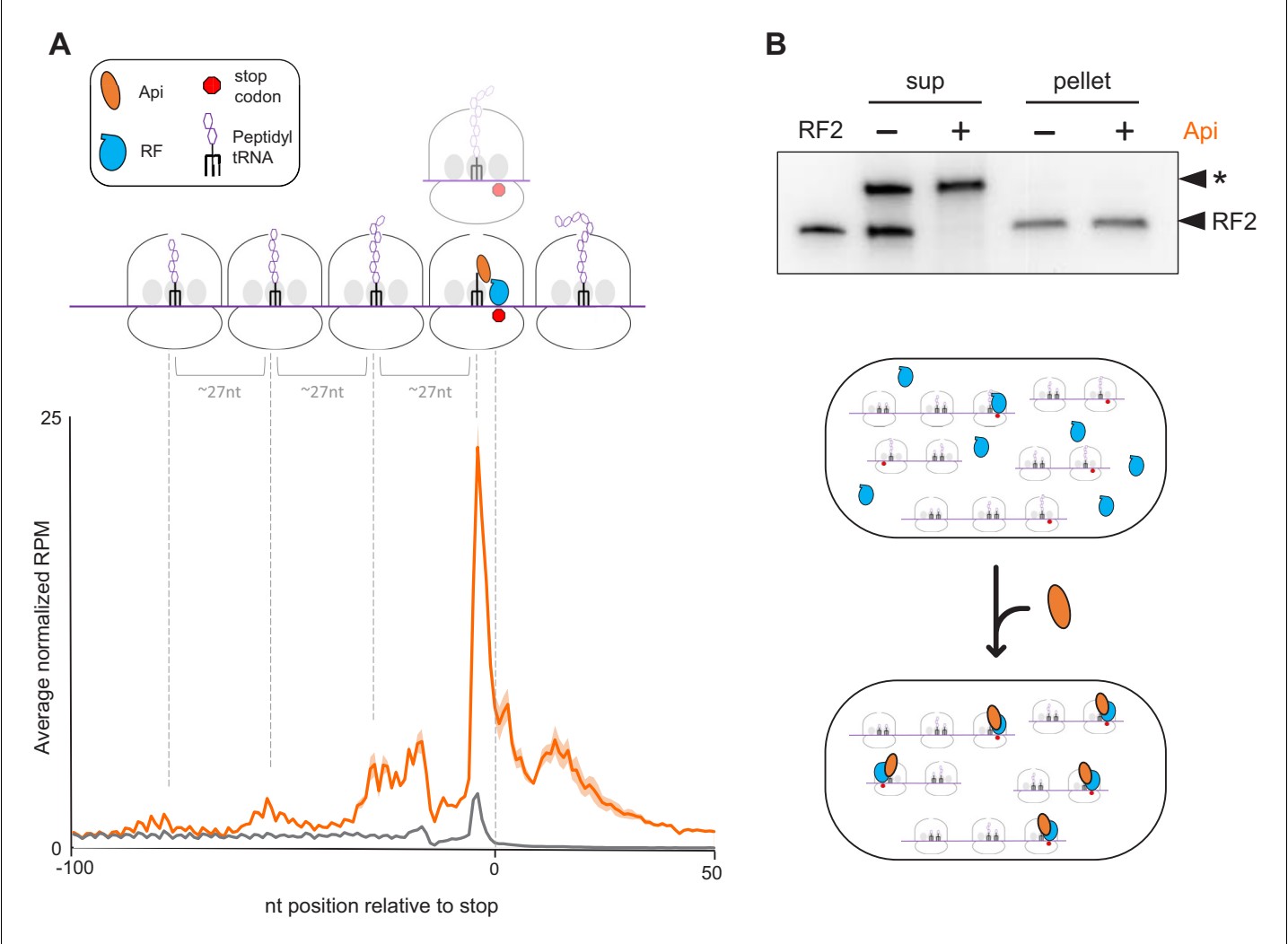

**Figure 2.** Api treatment leads to the accumulation of translating ribosomes near stop codons and depletion of free RFs in the cell. (**A**) Metagene plot of the average normalized ribosome occupancy near the annotated stop codons of genes in cells treated (orange) or not (gray) with Api. The solid line indicates the mean value and the shadow plot reflects the standard error between two independent biological replicates. The metagene plot is based on 836 actively translated genes separated from neighboring genes by ≥50 nt. The cartoon illustrates ribosomes stalled at the stop codon in the post- or pre-release state, ribosomes queued upstream from those stalled at termination sites, and ribosomes present in mRNA regions downstream from the main ORF stop codon (readthrough). (**B**) Analysis of ribosome-associated and free RF2 in untreated cells and cells treated with Api. Ribosomes from lysates of untreated or Api-treated cells were pelleted through a sucrose cushion. The presence of RF2 in the ribosome pellet or in the post-ribosomal supernatant was tested by western blotting using the anti-RF2 serum. The purified *E. coli* His$_6$-tagged RF2 was used as a control (lane marked 'RF2'). An unknown protein (indicated with an asterisk) cross-reacting with the anti-RF2 serum served as the loading control. The cartoon illustrates how Api depletes free RFs by sequestering them on the ribosome.

The online version of this article includes the following figure supplement(s) for figure 2:

**Figure supplement 1.** Footprint length distributions for Api and control samples.

**Figure supplement 2.** Puromycin treatment simplifies the ribosome distribution pattern in Api-treated cells.

**Figure supplement 3.** Api moderately increases ribosome density at start codons.

distance corresponding to the peak of the footprint length distribution (*Figure 2—figure supplement 1*), suggesting that this upstream density likely represents elongating ribosomes queued behind those arrested at stop codons. Even though ribosome collisions have been observed in bacteria and eukaryotes exposed to translation elongation inhibitors (*Kearse et al., 2019*; *Mohammad et al., 2019*), the extent of ribosome queuing in the Api-treated cells is especially pronounced. Because the queued ribosomes occupying inner codons of the ORFs carry nascent

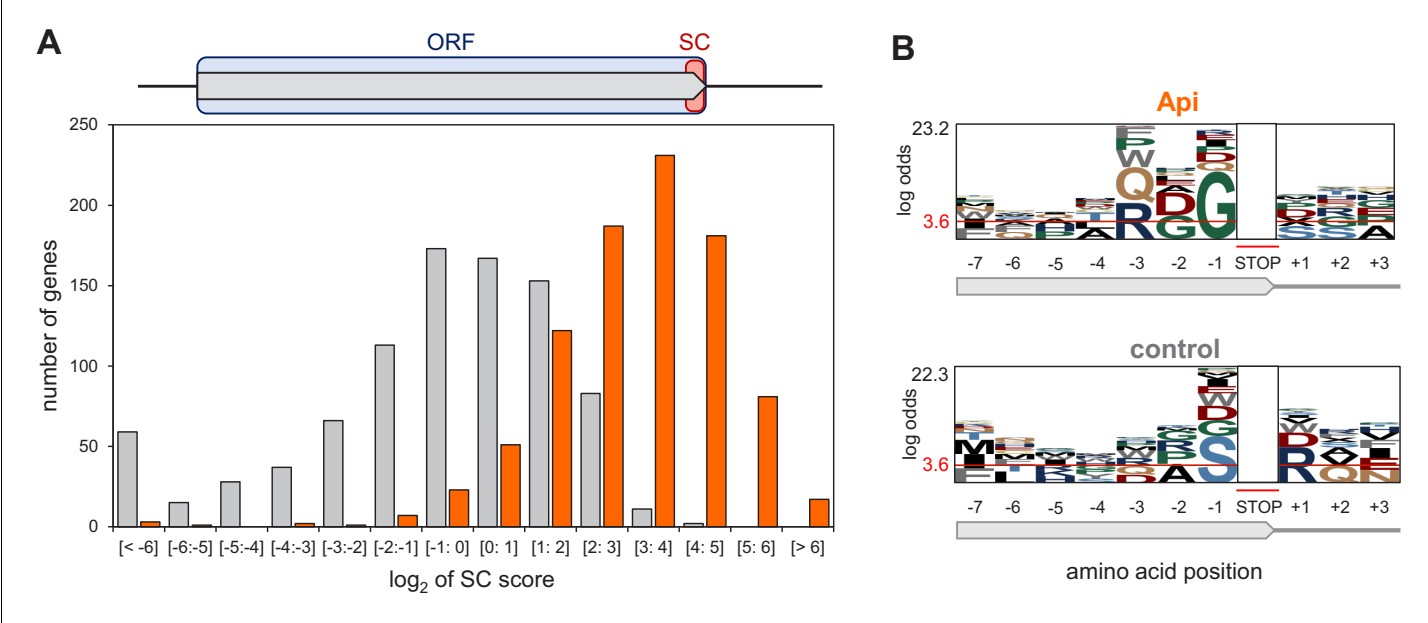

**Figure 3.** Api arrests ribosomes at translation termination sites. (**A**) Top: Stop codon (SC) score was calculated for actively translated genes separated by ≥50 nt as the ratio of ribosome density near the stop codon ('SC', orange rectangle) to the ribosome density within the entire ORF (light blue rectangle) (see Materials and methods for details). Bottom: Histogram of the SC scores of genes from cells treated (orange) or not (gray) with Api. (**B**) pLogo analyses of amino acid sequence bias around the termination regions of the top 50% SC-scoring versus the bottom 50% SC-scoring genes from Api-treated cells (top) or control cells (bottom).

The online version of this article includes the following figure supplement(s) for figure 3:

**Figure supplement 1.** Reproducibility of stop codon effects.

**Figure supplement 2.** Stop codon scores at different stop codons.

**Figure supplement 3.** Evaluation of cellular factors that could potentially contribute to Api-mediated ribosome arrest at stop codons.

polypeptides they should be refractory to Api binding (*Florin et al., 2017*). Yet, despite being Api-free, they are unable to complete the synthesis of the encoded protein and participate in new rounds of translation.

## Api treatment leads to pervasive stop codon readthrough

The metagene profile of the Api sample shows a dramatic increase in the number of ribosome footprints mapping to the regions downstream from the stop codons of the well-separated genes (*Figure 2A*). Similar to the SC score, we created the readthrough (RT) score to quantify the ribosome density within the first 40 nt of the 3'UTR relative to the ORF-associated density. In the control sample, the RT score of most genes is very low because few ribosomes are found downstream of stop codons. In contrast, in the Api sample, 90% of the genes have an RT score greater than 1, indicating higher relative density in the 3'UTR than in the ORF itself due to pervasive and efficient stop codon bypass (*Figure 4A*, *Figure 4—figure supplement 1*). We found no strong correlation between the stop codon identity and the RT score, indicating that all the stop codons are bypassed with comparable efficiency in the Api-treated cells, in contrast to the control cells, where UAA genes have significantly lower RT scores (*Figure 4A*, *Figure 4—figure supplement 2*).

The ribosome footprints in the 3'UTRs could represent post-termination ribosomes that have released the completed protein but somehow remained associated with mRNA and migrated past the stop codon. Alternatively, the 3'UTR footprints may correspond to ribosomes that failed to release the completed protein at the stop codon, bypassed it, and continued translation into the downstream segment of mRNA. Such a scenario could be caused by the depletion of available RF1/RF2 trapped by Api on terminating ribosomes. To test the feasibility of the latter model, we examined to which extent Api treatment affects the abundance of free RFs, more specifically of RF2, which by far outnumbers RF1 in the *E. coli* cytoplasm (*Schmidt et al., 2016*). Strikingly, unlike those

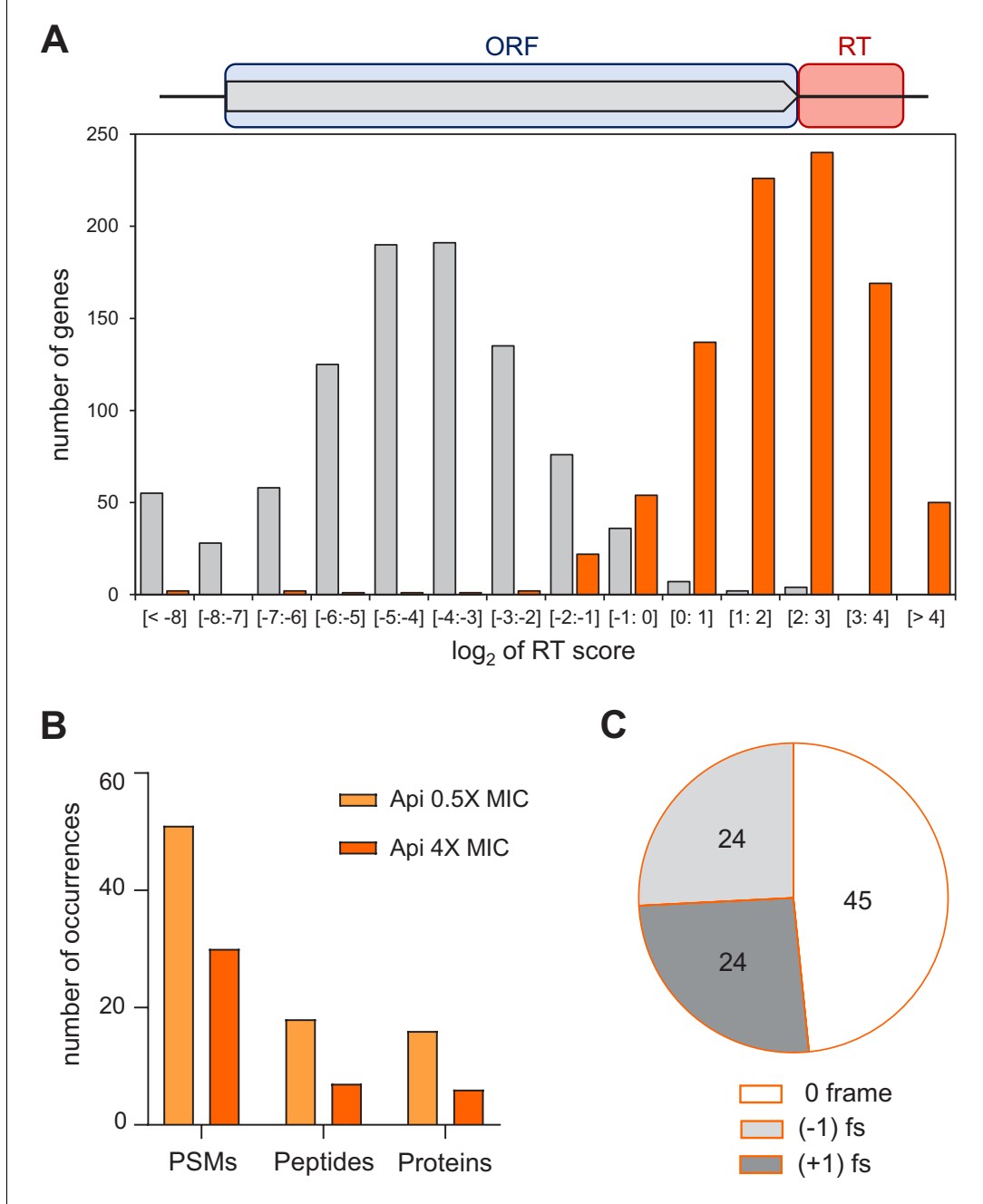

**Figure 4.** Api leads to stop codon bypass and generation of proteins with unintended C-terminal extensions. (**A**) Top: Readthrough (RT) score was calculated for actively translated genes separated by ≥50 nt as the ratio of ribosome density within the 40 nt downstream from the stop codon ('RT', orange rectangle) to the ribosome density within the associated ORF (light blue rectangle). Bottom: Histogram of the RT scores of genes from cells treated (orange) or not (gray) with Api. (**B**) The number of proteins with C-terminal extensions, peptides belonging to the C-terminal extensions of proteins, and peptide-spectrum matches (PSMs) identified by mass spectrometry in cells treated with 0.5× MIC (light orange bars) or 4× MIC (orange bars) of Api. With the same data-filtering criteria, no such products were detected in the control cells. (**C**) The number of peptides identified using relaxed criteria (see **Figure 4—figure supplement 3B**) in cells treated with 0.5× MIC of Api belonging to C-terminal extensions of proteins generated via bypassing the stop codon of the main ORF by amino acid misincorporation (0-frame), or as a result of −1 or +1 ribosomal frameshifting (fs). The online version of this article includes the following figure supplement(s) for figure 4:

**Figure supplement 1.** Reproducibility of stop codon bypass effects.
**Figure supplement 2.** Readthrough scores at different stop codons.
**Figure supplement 3.** In the Api-treated cells, ribosomes translate mRNA segments downstream from the stop codon of the main ORF.
**Figure supplement 4.** Stop codon bypass by near-cognate aminoacyl-tRNA misincorporation.

*Figure 4 continued on next page*

*Figure 4 continued*

**Figure supplement 5.** Possible scenarios for stop codon bypass via frameshifting in Api-treated cells.

**Figure supplement 6.** Api treatment does not lead to pronounced protein aggregation in *E*.

of the control sample, the ribosome-depleted lysates of Api-treated cells essentially lacked free RF2 (*Figure 2B*). In addition, we reasoned that if 3'UTR-associated ribosomes are engaged in translation they should preferentially stall at the downstream stop codons. Indeed, metagene analysis shows a peak of ribosome footprints at the first in-frame downstream stop codon (*Figure 4—figure supplement 3A*), indicating that ribosomes that bypassed the termination signal of the main ORF are actively translating downstream mRNA sequences.

We used shotgun proteomics to test whether exposure of cells to Api leads to productive translation of the non-coding mRNA segments and generation of proteins with C-terminal extensions. To allow for the accumulation of proteins synthesized during Api treatment, we exposed cells for 1 hr to a sublethal concentration of Api ($0.5\times$ MIC) permitting protein synthesis to continue, even if at a reduced level. In addition, we also incubated cells for 1 hr with a higher Api concentration ($4\times$ MIC). We then analyzed the proteins synthesized in treated and control cells, using custom algorithms to search for tryptic peptides encoded in the 150 nt-long segments downstream from the stop codons of the annotated ORFs (see Materials and methods). Even when applying strict filtering criteria (*Gessulat et al., 2019*), we were able to identify a significant number of peptides encoded in the genomic regions downstream from the ORF stop codons in both of the Api samples (*Figure 4B* and *Source data 3*). The same analysis found no such peptides in the control sample. The number of identified peptides encoded in the 3'UTRs increased further upon applying a less strict filter (*Figure 4—figure supplement 3B*). These results argued that genome regions downstream of the stop codons of the annotated ORFs are actively translated in the cells exposed to Api. Approximately half (49%) of the identified 'extension' peptides were encoded in frame with the upstream ORF (*Figure 4C*) and thus, were likely produced by the ribosomes that misread the main ORF stop codon as a sense codon and continued translating the downstream mRNA sequence. Consistently, in the three identified tryptic peptides encoded in mRNA sequences spanning the ORF termination signal, the stop codon was mistranslated by a near-cognate aminoacyl-tRNA (*Figure 4—figure supplement 4*). The other detected downstream peptides were encoded in the (−1) or (+1) frames relative to the main ORF (*Figure 4C* and *Source data 3*). These peptides also likely belong to C-terminal extensions of the main-ORF proteins given that stop codons can be bypassed via frameshifting of the ribosomes stalled at termination sites in the pre-release state, as was observed in the experiments of Gross and coworkers (*Baggett et al., 2017*). The sample size of the 'extension' peptides was insufficient to systematically identify the mRNA or nascent protein features conducing to stop codon bypass via frameshifting. However, the presence of a slippery sequence preceding the stop codon (e.g. in *arcB*; *Figure 4—figure supplement 5A*) or the possibility of repairing of peptidyl-tRNA in a different frame (e.g. in *rplE*; *Figure 4—figure supplement 5B*) could be among the factors contributing to the synthesis of identified proteins' C-terminal extensions.

Taken together, our results show that stop codon bypass, active translation of 3' UTRs, and production of proteins with C-terminal extensions constitute major consequences of Api action.

## Puromycin alters the Api stalled ribosome pattern

The results of our analysis suggest that the complex metagene profile observed in the vicinity of stop codons in the Api-treated cells is accounted for by distinct populations of ribosomes: the post-release ribosomes in complex with RF1/RF2 arrested by Api at the stop codons of the main ORF and the ribosomes carrying peptidyl-tRNA (including the queued ribosomes, some of the ribosomes in 3' UTRs, and a fraction of ribosomes stalled at the stop codons in pre-release state). We hypothesized that this complex profile could be at least partially simplified by clearing from mRNA the ribosomes carrying peptidyl-tRNA. For that, we used puromycin (Pmn), an antibiotic that releases the nascent protein from ribosomes whose PTC A site is accessible (*Traut and Monro, 1964*). Indeed, metagene analysis showed that Pmn treatment of lysate from Api-treated cells significantly reduced the relative amounts of queued ribosomes and 3'UTR ribosomes and, consequently, increased the relative occupancy of stop codons (*Figure 2—figure supplement 2*). Importantly, this does not imply that Pmn

causes more ribosomes to associate with stop codons but that a higher fraction of the remaining mRNA-bound ribosomes are positioned at termination sites. However, even upon Pmn treatment, some ribosomes remained within the ORFs, suggesting that either some of the translation complexes are refractory to Pmn action or that Pmn treatment requires further optimization. The remaining footprints still found in 3'UTRs after Pmn treatment may represent the ribosomes arrested at the downstream stop codons: Api-trapped RFs would block Pmn binding.

## Api treatment distorts the cell proteome and activates cellular ribosome rescue systems

Exposure of cells to Api leads to significant deregulation of the proteome. One of the most prominent effects is an increase in the relative content of ribosomal proteins and other factors related to the function and assembly of ribosomes (*Figure 5-figure supplement 1A* and *Source data 4*). Among the top 50 proteins whose abundance is increased in the cells exposed to 0.5× MIC of Api, 19 are related to translation (*Figure 5—figure supplement 1B*). This shift may reflect an attempt to compensate for the reduced protein synthesis capacity caused by the Api inhibitory action (*Source data 4*). Curiously, 13 other proteins among the top 50 are either associated with the inner or outer membrane or reside in the periplasm (*Figure 5—figure supplement 1B*), suggesting a still unknown link between Api action and protein secretion.

Analysis of the Ribo-seq data showed that the highest peak of ribosome occupancy in the Api-treated cells corresponds to the stop codon of the degron-encoding sequence in tmRNA (*Figure 5A,B*). tmRNA is a component of a conserved bacterial system that rescues stalled mRNA-associated ribosomes with an empty A site (*Buskirk and Green, 2017*; *Moore and Sauer, 2005*). Following tmRNA binding to the A-site, the ribosome switches templates, synthesizes the tmRNA-encoded degron sequence, and terminates translation, using class 1 RFs, at the UAA stop codon (*Buskirk and Green, 2017*; *Moore and Sauer, 2005*). tmRNA is known to operate on ribosomes that reach the 3' end of non-stop mRNAs or, potentially, those stalled due to scarcity of A-site ligand (*Ivanova et al., 2004*; *Janssen et al., 2013*; *Li et al., 2007*; *Subramaniam et al., 2014*). Both types of tmRNA substrates are generated in the cells treated with Api. First, the Api-stimulated stop codon bypass enables the translating ribosomes to approach the 3'-termini of RNA transcripts (*Figure 5C,D*), where they could be recognized as requiring rescue. In addition, ribosomes stalled at stop codons in a pre-release state, and possibly the queued ribosomes as well, whose A site is vacant, could serve as tmRNA substrates.

The sharp increase of ribosome footprints at the tmRNA stop codons in the Api samples (*Figure 5B*) argues in favor of the massive recruitment of tmRNAs to the stalled ribosomes. However, due to the Api-mediated depletion of the available RFs (*Figure 2B*), the degron-tagged protein cannot be released, thereby preventing proper tmRNA function and recycling. Consistently, the cell attempts to activate the two other *E. coli* backup rescue factors, ArfA and ArfB (reviewed in *Buskirk and Green, 2017*). For the *arfA* gene in the Api-treated cells, we observed a nearly tenfold increase in the abundance of its transcript and a more than sevenfold higher number of ribosomes associated with its mRNA (*Figure 5D* and *Source datas 5* and *6*). The relative amount of *arfB* transcripts increases by ~40% and there are ~4 times more ribosomes per *arfB* transcript (*Source datas 5* and *6*). However, because ribosome rescue by tmRNA or *arfA* requires the availability of RFs and because ArfB, which acts as a termination factor itself, becomes trapped in the ribosome by Api (*Chan et al., 2020*), it is unlikely that either of the rescue systems can mitigate the consequences of Api action. Consistently, deletion of tmRNA, *arfA*, or *arfB* genes does not increase cell sensitivity to Api (*Source data 7*).

## Api increases start codon occupancy at some genes

While the predominant effect of Api on cellular translation is the build-up of ribosome density in the vicinity of stop codons (*Figure 2A*), a metagene analysis of the 5' ends of genes showed a moderate increase (~twofold) of the number of ribosome footprints at the start codons of the ORFs in the Api sample compared to the control (*Figure 2—figure supplement 3A*). Among the genes with the strongest start codon effects, we selected *zntA*, *srmB*, *cysS*, *tyrS*, and *yhbL* to test in vitro whether Api directly delays the ribosome departure from the initiator codon. While translation arrest at stop codons is readily detected by toeprinting analysis at 50 µM of Api (*Florin et al., 2017*), no start

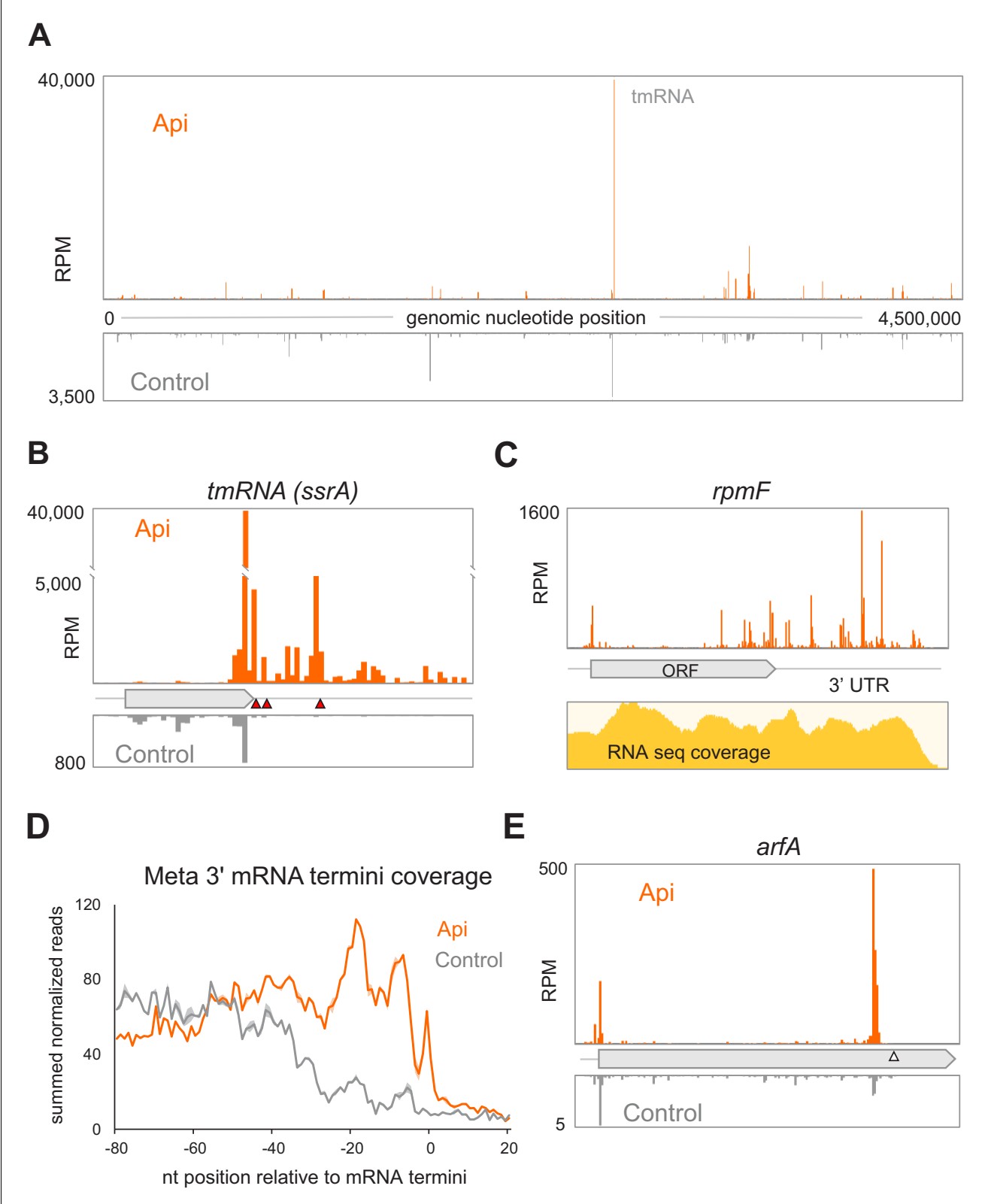

**Figure 5.** Alteration of the proteome and activation of the ribosome rescue systems in response to Api. (**A**) The density of ribosome footprints throughout the *E. coli* genome in cells treated or not with Api. (**B**) The density of ribosomal footprints at the termination region of the tmRNA ORF encoding the degron tag. The position of the main stop codon and two in-frame downstream stop codons of the tmRNA are indicated with red triangles. (**C**) Ribosome footprints near the 3'-ends of the *rpmF* transcript in Api-treated cells. The RNA-seq coverage is shown below. (**D**)

*Figure 5 continued on next page*

*Figure 5 continued*

The Metagene plot of summed normalized reads toward the 3′ termini of 1006 mRNAs (***Dar and Sorek, 2018***; see Materials and methods). (**E**) The density of ribosome footprints in the *arfA* gene in Api-treated or control cells. The gray triangle marks the position of the RNase III cleavage site involved in the tmRNA- and ArfA-dependent regulation of *arfA* expression.

The online version of this article includes the following figure supplement(s) for figure 5:

**Figure supplement 1.** Api-induced changes in protein abundance.

---

codon effects were observed at this concentration of the inhibitor in any of the tested ORFs. However, increasing Api concentration to 2 mM led to the appearance of a start codon toeprint bands in the *zntA* and *tyrS* templates (***Figure 2—figure supplement 3B–C***); the start codon effects were much weaker or completely absent with the other tested genes. The moderate effect of Api on translation initiation is reminiscent of the main mode of action of other PrAMPs (***Gagnon et al., 2016***; ***Roy et al., 2015***; ***Seefeldt et al., 2016***; ***Seefeldt et al., 2015***) and could be explained by its ability to bind with a reduced affinity to the ribosome even in the absence of RF1/RF2 (***Florin et al., 2017***; ***Kolano et al., 2020***; ***Krizsan et al., 2014***). However, it remains to be elucidated whether the mechanism of the weak inhibition of translation initiation by Api is comparable to that of other PrAMPs.

## Discussion

Ribo-seq and proteomics studies revealed a complex pattern of events triggered by the translation termination inhibitor Api in bacteria.

One of the main effects of Api in the cell is the arrest of translation at the stop codons of the ORFs. A larger fraction of the ribosomes associated with stop codons is likely arrested in a post-release state, with Api bound in the exit tunnel, trapped RF1 or RF2 in the A site, and deacylated tRNA in the P site. Most of the available RFs are sequestered in these complexes (***Figure 2B***), which explains why overexpression of class 1 RFs reduced the sensitivity of *E. coli* to Api (***Matsumoto et al., 2017***). In normal conditions, however, because the number of ribosomes in the cell exceeds the copy number of RF1 and RF2 (***Schmidt et al., 2016***), some ribosomes likely become stalled at stop codons in a pre-release state, with a vacant A site and the completed protein still esterifying P-site tRNA. It is difficult to determine experimentally what fraction of the stop codon associated footprints belongs to the pre-release ribosomes. However, because the number of the total stop codon footprints (~11% of all mapped footprints) roughly matches the ratio between ribosomes and RFs in the cell (***Schmidt et al., 2016***), it is likely that most of the ribosome density detected at stop codons is accounted for by Api-stalled post-release ribosomes. A fraction of the stop codon peak could be also possibly attributed to the post-release Api-free ribosomes that have not yet been recycled (***Schuller et al., 2017***).

The extent of Api-induced ribosome arrest at stop codons differs in magnitude between genes. At least some of the gene- or nascent protein-specific effects of the Api action could be linked to the kinetics of extrusion of the nascent chain from the exit tunnel, which is a pre-requisite for Api binding. The nascent chains that would linger for a longer time after their separation from the P-site tRNA would allow more time for RF1/RF2 dissociation before Api reaching its binding site in the exit tunnel. Consistently, we note a modest trend for arrest at the RF2-specific UGA codons to be somewhat less pronounced than that at the UAA or UAG stop codons (***Figure 3—figure supplement 2***) possibly because the residence time of RF2 on the post-release ribosome could be shorter than that of RF1 (***Adio et al., 2018***), which would provide Api with a lesser chance to trap the ribosome terminating at the UGA codons. We should note, however, that the available data are insufficient to distinguish whether the observed context effects operate upon the post- or pre-release fractions of the stop codon-associated ribosomes.

Api-induced ribosome stalling at stop codons causes severe ribosome queuing. The impressive ability of Api to cause queuing likely stems from its unique mechanism of action. In contrast to elongation inhibitors, which simultaneously arrest ribosomes at many codons within the gene, Api preferentially arrests translation at a single site, the stop codon; the trailing ribosomes on the same mRNA continue translation until they run into the traffic jam caused by the termination site roadblock. The

pronounced ribosome queuing in the Api-treated cells illustrates a distinctive feature of a translation termination inhibitor as an antimicrobial: Api needs to bind to only a few terminating ribosomes to preclude a much larger number of inhibitor-free ribosomes from participating in translation. Termination arrest and ribosome queuing, however, did not increase the number of heavy polysomes in the cell, likely because in fast-growing cells ~90% of the ribosomes are already associated with mRNAs (*Bartholomäus et al., 2016*; *Bremer and Dennis, 1996*) and thus, blocking termination would not perceptibly increase ribosome loading per mRNA.

Our Ribo-seq experiments revealed a striking and pervasive Api-induced genome-wide stop codon readthrough resulting in an increased abundance of ribosome footprints downstream of ~99% of the analyzed genes. The appearance of peptides encoded in the downstream mRNA sequences, revealed by shotgun proteomics, and the increased ribosome occupancy of the downstream stop codons strongly argue that in the Api-treated cells a sizable fraction of ribosomes translate past the end of the ORFs and synthesize proteins with C-terminal extensions. Stop codon bypass is likely a consequence of the Api-mediated depletion of available RFs (*Figure 2B*) that increases the residence time of pre-release state ribosomes at the stop codons of the ORFs, leading ultimately to readthrough.

Some level of stop codon readthrough can be also induced by antibiotics that render the ribosome error-prone, such as aminoglycosides, odilorhabdins, or negamycin (*Olivier et al., 2014*; *Pantel et al., 2018*; *Polikanov et al., 2014*; *Thompson et al., 2004*; *Wangen and Green, 2020*). However, Api's ability to stimulate bypass appears to be more robust, leading to the appearance of ribosome footprints downstream of not only the first but also of the subsequent stop codons (*Figure 4—figure supplement 3A*). Because Api does not inhibit translation elongation (in contrast to the aforementioned antibiotics; *Cabañas et al., 1978*; *Polikanov et al., 2014*; *Wang et al., 2012*), more ribosomes can reach the ends of the ORFs in the Api-treated cells, thereby triggering a higher rate of stop codon readthrough. Additionally, unlike miscoding antibiotics, Api does not reduce the general accuracy of translation and thus, could be exploited for medical applications where a premature stop codon readthrough is desirable (*Huang et al., 2019*; *Keeling et al., 2014*). Api-mediated stop codon suppression can be also instrumental in synthetic biology, for example, for enhancing the incorporation of non-canonical amino acids (*Florin et al., 2017*; *Kuru et al., 2020*).

Api-induced stop codon bypass is achieved by either misincorporation of a near-cognate amino acid at the stop codon or via (−1) or (+1) frameshifting leading to the synthesis of proteins with C-terminal extensions. Although under our Api treatment conditions, we did not detect any significant accumulation of protein aggregates (*Figure 4—figure supplement 6*), it remains possible that accumulation of proteins with C-terminal appendages could contribute to the antimicrobial activity of Api since these aberrant extensions may negatively affect the proteins' catalytic capacity, ligand binding, or interactions with other polypeptides.

By allowing translation to proceed to the 3' ends of mRNA transcripts (*Figure 5C,D*) and by prompting some of the translating ribosomes to stall at the stop codons in a pre-release state, Api triggers the activation of the ribosome rescue systems. However, in the cells exposed to Api, the rescue systems are likely worthless. tmRNA-mediated rescue of stalled ribosomes relies on canonical translation termination at the UAA codon of the degron ORF in tmRNA (reviewed in *Buskirk and Green, 2017*). Yet, the lack of available RF1/RF2 in the Api-treated cells prevents the release of the degron-tagged protein, therefore sequestering tmRNA. The depletion of the available tmRNA pool leads to activation of the backup rescue systems, which nevertheless, should be also ineffectual in the Api-treated cells. ArfA facilitates translation termination at non-stop mRNAs by interacting with the empty A site of the small ribosomal subunit and promoting binding of RF2, which can then release the stalled protein. Once again, the lack of available RF2 in the Api-treated cells should prevent ArfA from carrying out its rescue mission. This idea is supported by the appearance of an exceedingly high ribosome density peak at the end of the non-stop *arfA* mRNA (*Figure 5E*), where the stalled ribosomes are expected to be released by the action of the ArfA protein itself (*Garza-Sánchez et al., 2011*). Despite acting independently from the primary RFs, the alternative RF ArfB, should be equally inconsequential because Api can also trap ArfB on the ribosome (*Chan et al., 2020*) and deplete its available pool. Thus, it appears that none of the known ways that the bacterial cell normally exploits for salvaging stalled ribosomes would be of much use when the translation is inhibited by Api. In agreement with this notion, the inactivation of any of the rescue systems does not sensitize *E. coli* to the Api action (*Source data 7*).

One of the unexpected aspects of Api action revealed by the in vivo studies is the increased ribosome occupancy of start codons of some genes. These effects could be related to the reported low affinity of Api for the ribosome with the vacant nascent peptide exit tunnel (*Florin et al., 2017*; *Kolano et al., 2020*; *Krizsan et al., 2014*). Once associated with the ribosome at the start codon, Api may block the first peptide bond formation by displacing the fMet moiety of the initiator tRNA in the P site or may interfere with the first act of translation. Because the initiation effects could be observed in vitro only at very high concentrations of Api (*Figure 2—figure supplement 3B–C*), the Api-induced ribosome stalling at start codons in the cell could involve additional factors absent in the cell-free system or result from the accumulation of aberrant ribosomes.

On the basis of our findings, we can propose a model of Api action in the bacterial cell involving multiple, not necessarily sequential effects (numbered *1* to *5* in *Figure 6*). The immediate effect of Api is arresting the ribosomes in the post-release state at stop codons and sequestering the available RF1/RF2 (*1*). Because ribosomes outnumber RFs, the depletion of available RFs leads to stalling of a fraction of the translating ribosomes at the stop codons in the pre-release state, with peptidyl-tRNA bound in the P-site and an empty A site (*2*). The trailing ribosomes that carry incomplete polypeptides form a stalled queue behind those trapped at the stop codons (*3a,3b*). A prolonged arrest of pre-release ribosomes at termination leads to the eventual bypass of stop codons via misincorporation of a near-cognate tRNA or frameshifting resulting in the synthesis of proteins with C-terminal extensions (*4*). Some of the ribosomes that bypassed the main ORF stop codon may ultimately reach

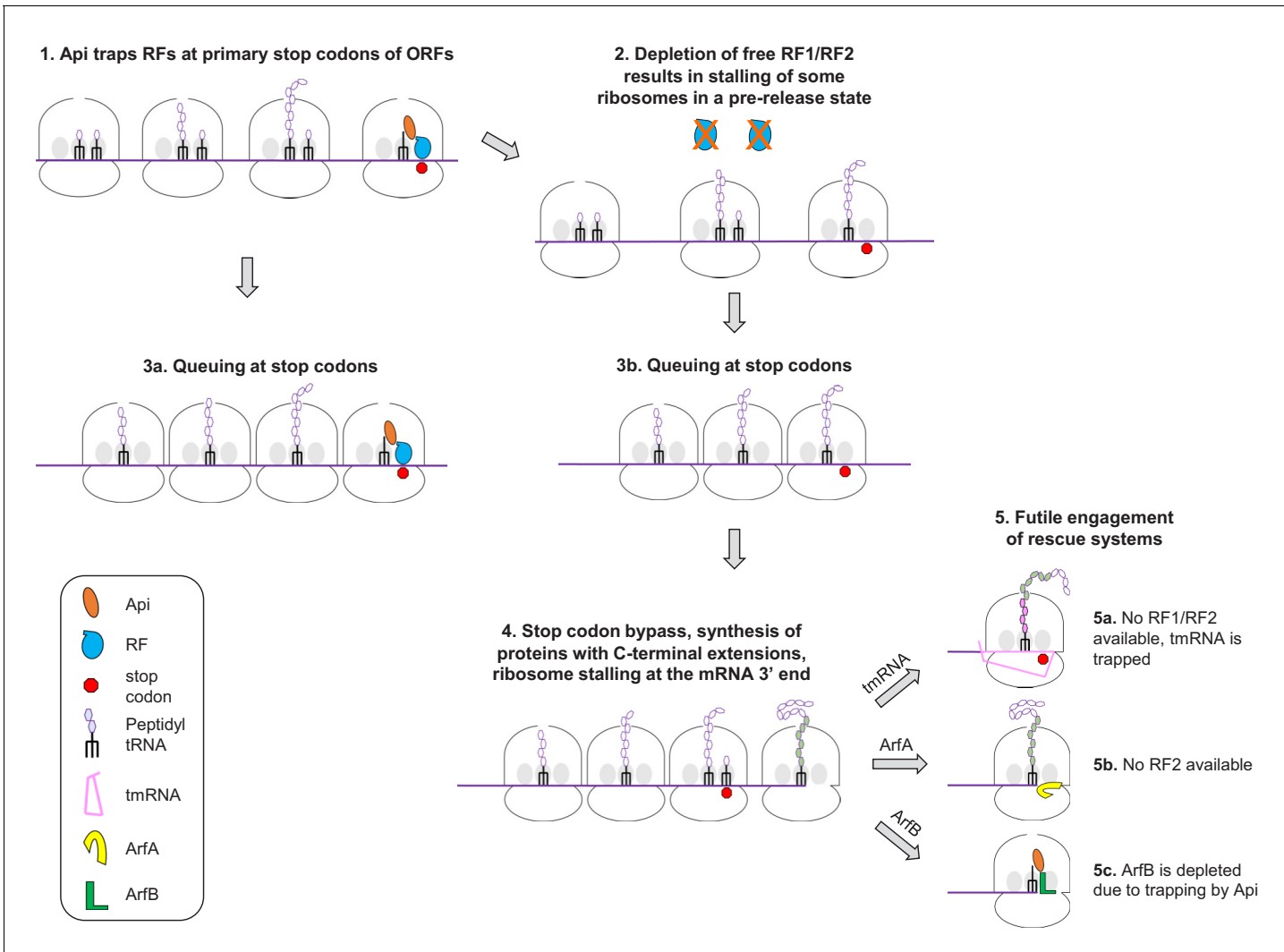

**Figure 6.** A model of Api action in the bacterial cell.

the mRNA 3′ end. Accumulation of ribosomes with vacant A-site (stalled at the 3′ ends of the RNA transcripts or at the stop codons) triggers the massive but futile engagement of the ribosome rescue systems (5). The combination of these effects causes rapid and efficient cessation of translation.

Understanding the mode of Api action in the bacterial cell helps to rationalize the seeming discrepancy between its marginal effect on reporter expression in a cell-free system and the strong inhibition of translation in vivo (*Castle et al., 1999*; *Krizsan et al., 2015*; *Krizsan et al., 2014*). Exposure of cells to Api, leads to translation arrest at stop codons of multiple ORFs, ribosome queuing and depletion of available RFs resulting in a rapid halt of protein synthesis. By contrast, in cell-free systems, where only one type of mRNA is present, many mRNA molecules will be translated at least once before the Api-induced depletion of the RF pool, thereby resulting in poor inhibition of the reporter expression.

The use of Ribo-seq in combination with the translation initiation inhibitor retapamulin was instrumental in identifying translation start sites in bacteria, revealing a universe of cryptic protein-coding sequences (*Meydan et al., 2019*; *Weaver et al., 2019*). Ribo-seq, enhanced by the use of a translation termination inhibitor, could bolster our ability to detect the translated sequences in the bacterial genomes. Because the effects of Api are multifaceted and lead not only to translation arrest at the stop codons but also to ribosome queuing and stop codon readthrough (*Figure 2A*), its immediate application for mapping translation termination sites could be challenging. Nevertheless, with additional tricks, for example, properly optimized Pmn treatment to remove queued and elongating ribosomes (*Figure 2—figure supplement 2*), Api could be repurposed for genome-wide analysis of stop codons of protein-coding regions. The use of Ribo-seq in combination with inhibitors of translation initiation and termination could provide new insights into the principles of genetic encoding and regulation of gene expression.

## Materials and methods

**Key resources table**

| Reagent type (species) or resource | Designation | Source or reference | Identifiers | Additional information |
|---|---|---|---|---|
| Strain, strain background (*Escherichia coli*) | BL21 (Δ*tolC*) | *Meydan et al., 2019* | | |
| Strain, strain background (*Escherichia coli*) | BW25113 (Δ*arfA*) | *Baba et al., 2006* | | |
| Strain, strain background (*Escherichia coli*) | BW25113 (Δ*arfB*) | *Baba et al., 2006* | | |
| Strain, strain background (*Escherichia coli*) | BW25113 (Δ*smpB*) | *Baba et al., 2006* | | |
| Strain, strain background (*Escherichia coli*) | BW25113 | *Baba et al., 2006* | | |
| Antibody | Anti-RF2 (rabbit serum polyclonal) | Cruz-Vera laboratory | | (1:500) |
| Sequence-based reagent | zntA_F | Integrated DNA Technologies | PCR primers | TAATACGACTCACTATAGGGAAA CTTAACCGGAGGATGCCATGTCG |
| Sequence-based reagent | zntA_R | Integrated DNA Technologies | PCR primers | GGTTATAATGAATTTTGCTTATTA ACTTTGAACGCAGCAAATTGAGGGGC |
| Sequence-based reagent | tyrS_F | Integrated DNA Technologies | PCR primers | TAATACGACTCACTATAGGGTTATA TACATGGAGATTTTGATGGCAAGC |
| Sequence-based reagent | tyrS_R | Integrated DNA Technologies | PCR primers | GGTTATAATGAATTTTGCTTATTAA CGACTGGGCTACCAGCCCCCG |
| Sequence-based reagent | Linker oligonucleotide: NI-810-NI817 | *McGlincy and Ingolia, 2017* | | /5Phos/NNNNNATCGTAGA TCGGAAGAGCACACGTCTGAA/3ddC/ |

*Continued on next page*

*Continued*

| Reagent type (species) or resource | Designation | Source or reference | Identifiers | Additional information |
|---|---|---|---|---|
| Sequence-based reagent | RT primer | Integrated DNA Technologies | | /5Phos/RNAGATCGGAAGAGCGT CGTGTAGGGAAAGAG /iSp18/GTGACTGGAGTTCAGACGTGTGCTC |
| Peptide, recombinant protein | Api137 | NovoPro Biosciences Inc | | |
| Commercial assay or kit | Oligo Clean and Concentrator kit | Zymo Research | D4060 | |
| Commercial assay or kit | MEGAclear | Thermo Fisher | AM1908 | |
| Commercial assay or kit | MicrobEXPRESS | Thermo Fisher | AM1905 | |
| Commercial assay or kit | High pH reverse phase peptide fractionation | Thermo Fisher | 84868 | |
| Chemical compound, drug | GMPPNP | Sigma | G0635 | |
| Chemical compound, drug | L-[35S]-methionine | Perkin Elmer | | |
| Peptide, recombinant protein | MNase (S7 Microccocal Nuclease) | Roche | 10107921001 | |
| Peptide, recombinant protein | DNase I, RNase free | Roche | 04716728001 | |
| Peptide, recombinant protein | SUPERase*In | Thermo Fisher | AM2696 | |
| Peptide, recombinant protein | T4 PNK | New England Biolabs | M0201L | |
| Peptide, recombinant protein | Mth RNA Ligase | New England Biolabs | E2610S | |
| Peptide, recombinant protein | 5' Deadenylase | New England Biolabs | M0331S | |
| Peptide, recombinant protein | RecJf | New England Biolabs | M0264S | |
| Peptide, recombinant protein | Superscript III | Invitrogen | 18080–093 | |
| Peptide, recombinant protein | CircLigase ssDNA Ligase | Lucigen | CL4115K | |
| Peptide, recombinant protein | Phusion high-fidelity polymerase | New England Biolabs | M0530S | |
| Peptide, recombinant protein | MS grade modified Trypsin | Promega | V5113 | |
| Software, algorithm | Scripts for RNA-seq and Ribo-seq metagene analysis | https://github.com/adam hockenberry/ribo-t-sequencing | | |
| Software, algorithm | Scripts for Ribo-seq processing | https://github.com/ mmaiensc/RiboSeq | | |
| Software, algorithm | Bedtools | https://github.com/ arq5x/bedtools2 | | |
| Software, algorithm | pLogo | https://plogo.uconn.edu/ | | |
| Software, algorithm | Bowtie2 | https://github.com/ BenLangmead/bowtie2 | v2.2.9 | |
| Software, algorithm | cutadapt | https://github.com /marcelm/cutadapt/ | | |

*Continued on next page*

Continued

| Reagent type (species) or resource | Designation | Source or reference | Identifiers | Additional information |
|---|---|---|---|---|
| Software, algorithm | Mass-spectrometry raw data conversion | *Chambers et al., 2012* | | |
| Software, algorithm | Peptide database searching | *Xu et al., 2015* | | |
| Software, algorithm | Peptide database searching | *Kong et al., 2017* | | |
| Software, algorithm | Custom databases used for peptide searching | http://git.pepchem.org/gaolab/api_ribo_ext | | |
| Software, algorithm | Data analysis scripts in Python | http://git.pepchem.org/gaolab/api_ribo_ext | | |

## Reagents

Api137 (*Berthold et al., 2013*), which we refer to simply as Api, was synthesized by NovoPro Biosciences Inc DNA oligonucleotides were synthesized by Integrated DNA Technologies.

## Protein synthesis inhibition assay

The inhibition of cellular protein synthesis by Api was analyzed by metabolic labeling as described previously (*Meydan et al., 2019*) with the following modifications. *E. coli* BL21 Δ*tolC* cells were grown overnight in MOPS medium (Teknova) lacking methionine (MOPSΔMet). Cells were diluted 1:200 into fresh MOPS-Met and grown at 37°C until the culture reached a density of $A_{660}$ of 0.2. Aliquots of the exponentially growing cells were added to tubes containing appropriately diluted Api in MOPSΔMet medium to obtain 0.75, 1.5, 3.1, 6.25, 12.5, 25, and 50 μM as final Api concentrations in the total volume of 100 μL. After 2 min incubation, 28 μL were transferred to another tube containing 2 μL MOPSΔMet medium supplemented with 0.3 μCi of L-[$^{35}$S]-methionine (specific activity 1,175 Ci/mmol; MP Biomedicals). Following 90 s incubation, the content was transferred onto Whatman 3 MM paper discs pre-wetted with 5% TCA and the procedure was continued as described previously (*Meydan et al., 2019*).

The time course of protein synthesis inhibition was performed as described above, except that Api was directly added to a final concentration of 6.25 μM to the exponentially growing culture in MOPSΔMet medium and aliquots were taken at 0, 2, 5, and 10 min.

## Cell growth and cell lysates preparation for Ribo-seq experiments

The Ribo-seq experiments were performed as described previously (*Meydan et al., 2020*; *Meydan et al., 2019*) using cells grown in MOPS medium. Briefly, the overnight cultures of *E. coli* BL21 Δ*tolC* cells were diluted 1:200 in 150 mL of medium and grown at 37°C in flasks until reaching density of $A_{600}$ ~0.4. For Api-samples, 4.12 mg of dry Api powder was added directly to the cultures to a final concentration of 4× MIC and incubation with shaking continued for 5 min. Untreated (control) and Api-treated cells were harvested by rapid filtration, flash frozen in liquid nitrogen and cryolysed in 650 μL lysis buffer (20 mM Tris pH 8.0, 10 mM $MgCl_2$, 100 mM $NH_4CL$, 5 mM $CaCl_2$, 0.4% Triton X-100, 0.1% NP-40) supplemented with 65 U RNase-free DNase I (Roche) and 208 U Superase•In RNase inhibitor (Invitrogen). The pulverized cells were thawed at 30°C for 2 min and incubated in an ice water bath for 20 min and lysates were clarified by centrifugation at 20,000 × g for 10 min at 4°C. One aliquot of the clarified lysates was preserved frozen for RNA-seq analysis, another aliquot was immediately treated with MNase (see below); the third aliquot was treated by puromycin (Pmn) as follows. The cell lysates were supplemented with 1 mM Pmn (Millipore-Sigma), 10 mM creatine phosphate (Millipore-Sigma) and 40 μg/mL creatine kinase (Roche), and incubated for 10 min at 25°C with shaking at 1400 rpm. Cell lysates, treated or not with Pmn, were subjected to MNase treatment by adding 450 U MNase (Roche) per 25 $A_{260}$ units of lysate, 3 mM GMPPNP, and incubating for 60 min at 25°C. The MNase reaction was quenched by the addition of EGTA (5 mM final concentration). Following sucrose gradient centrifugation and collection of the 70S ribosome peak (*Becker et al., 2013*), subsequent isolation of ribosomal footprints and library preparation was performed as described (*McGlincy and Ingolia, 2017*).

### Processing of ribosome footprints for Ribo-seq analysis

A custom script was used to demultiplex the samples, remove the linker barcode and any nts 3′ of the linker barcode, and then remove five nts from the 3′ end and 2 nt from the 5′ end, which were added as part of the library design (*McGlincy and Ingolia, 2017*; *Meydan et al., 2020*; *Meydan et al., 2019*). Bowtie2 (v2.2.9) within the GALAXY pipeline first aligned the trimmed reads to the non-coding RNA sequences. The remaining unmapped reads were aligned to the reference BL21 genome (GenBank ID CP001509.3; *Meydan et al., 2020*; *Meydan et al., 2019*). The 26 to 39 nt-long reads were used in the subsequent analyses. After analyzing the footprints at the start codons, the first position of the P-site codon was assigned 16 nt from the 3′ end of the read, in-line with the 15 nt offset previously proposed (*Mohammad et al., 2019*).

### Metagene analysis

The metagene analyses at the annotated start and stop regions followed the previously described protocol (*Aleksashin et al., 2019*). For the inclusion in the analysis, the well-separated ORFs had to satisfy three criteria in all four datasets (two control and two Api-treated biological replicates): (i) The ORF length is at least 300 nt, (ii) at least 5% of the positions within the coding sequence had assigned reads values above zero, and (iii) the average number of rpm per nt in an ORF is $\geq$0.005. Ribosome footprint density at each nucleotide was normalized to the average coverage of the entire ORF plus 50 nt up- and downstream. The mean of the normalized values was calculated and plotted for the regions around the start and stop codons of the ORF.

The metagene analysis of the first in-frame stop codons of the downstream regions was carried out using 3769 genes with such codons present within the 225 nt downstream of the annotated ORF. Normalized read coverage, calculated for each nucleotide within a 200 nt window centered around the last nucleotide of the downstream stop codons, was determined by dividing ribosome footprint coverage at each position by the maximum coverage value within the window. Normalized coverage was then summed for each position in the windows and plotted.

To analyze the metagene coverage at the 3′ termini of mRNAs, we first converted the 3′ termini coordinates reported in the *E. coli* BW25113 strain (*Dar and Sorek, 2018*) to coordinates in the BL21 strain, used in this work. We then followed a similar normalization and summing procedure to that used in the first in-frame stop codon analysis. However, the window of nts for normalization included 100 upstream and 20 downstream of the mRNA 3′ termini.

### Calculation of stop codon- and readthrough scores

The well-separated genes included in the analysis were required to meet two criteria in all four datasets (two control and two Api-treated biological replicates): (i) they were separated from the upstream and downstream adjacent genes by at least 50 nt, and (ii) they have a coverage density of at least 0.1 reads per nt in the protein-coding sequence, excluding the first and last 9 nt of the ORF. The stop codon (SC) region was assigned as nine 3′-terminal nt of the ORF (the last two sense codons and the stop codon); the readthrough (RT) region encompassed 40 nt downstream from the ORF stop codon. The SC and RT scores were calculated by dividing the ribosome density within the SC region or the RT region, respectively, by the total ORF density. The mean SC and RT scores of each gene were used in further analyses.

### RNA-seq analysis

Aliquots of the clarified cell lysates from the samples prepared for the Ribo-seq analysis (see above) were used to extract and analyze total RNA as previously described (*Aleksashin et al., 2019*). Briefly, total RNA was purified by acidic phenol extraction, short RNAs and rRNA were subtracted using the MEGAclear (Ambion) and MicroEXPRESS (Ambion) kits, respectively, mRNA was fragmented via alkaline fragmentation and, after size-fractionation and isolating fragment sin the 15–45 nt range, converted to the library for next-generation sequencing.

The RNA-seq analysis was performed as previously described (*Aleksashin et al., 2019*). Briefly, Cutadapt was used to process and demultiplex the reads. Kallisto (v.0.44; *Bray et al., 2016*) and Sleuth (v.0.30; *Pimentel et al., 2017*) were used to align the reads and perform a differential expression analysis, respectively. To align reads to the *ssrA* transcript, the GALAXY pipeline was used for alignment and quantifying reads per million after filtering out reads aligning to the non-coding RNA.

## Analysis of free RF2 abundance

*E. coli* BL21Δ*tolC* cells were grown in 100 mL of MOPS media to a density of $A_{600}$ ~0.5, split into separated flasks and either treated with 4× MIC of Api for 5 min or left untreated (control). The flasks were cooled in an ice water bath and cell cultures were centrifuged in a rotor JA-25.50 (Beckman) at 4°C and 4000× g for 10 min. The cell pellet was resuspended in 400 µL cold lysis buffer (20 mM Tris pH 8.0, 10 mM $MgCl_2$, 100 mM $NH_4Cl$, 5 mM $CaCl_2$, 0.4% Triton X-100, 0.1% NP-40), 10 U/µL RNAse free DNase I (Roche) and 20 U/µL Superase•In RNase inhibitor (Invitrogen) frozen in liquid nitrogen and cryo-milled. The pulverized cells were thawed at 30°C for 2 min, incubated in an ice water bath for 20 min and lysates were clarified by centrifugation for 10 min at 20,000 × g and 4°C. About 150 µL of the lysate was layered onto a 600 µL sucrose cushion (25% sucrose, 20 mM tris pH 8.0, 10 mM $MgCl_2$, 100 mM $NH_4Cl$) in a TLA100.2 rotor (Beckman) and centrifuged for 1 hr at 100,000 rpm and 4°C. The supernatant was collected, and the ribosome pellet was resuspended in buffer (20 mM Tris pH 8.0, 10 mM $MgCl_2$, 100 mM $NH_4Cl$, 1% SDS). Samples were separated by 4–20% SDS-PAGE gels, then electroblotted onto nitrocellulose membranes followed by Ponceau staining and probing with rabbit anti-RF2 antibody. Antibody binding was visualized using a secondary antibody (goat anti-rabbit IgG labeled with horseradish peroxidase) and imaged chemiluminescence.

## Sample preparation

### For proteomics analysis

All protein preparation steps were performed on ice. The cell pellet from each sample was resuspended in 2 mL TNI lysis buffer (50 mM Tris-HCl, pH 7.5, 250 mM NaCl, 0.5% Igepal CA-630, 1 mM EDTA, and one complete ULTRA EDTA-free Protease Inhibitor cocktail tablet), sonicated using a probe sonicator (Branson, CT) at 30% power at 55 kHz repeatedly for 20 s with 10 s interval to cool down the lysate until sufficiently clear lysate was obtained. The lysate was then clarified by centrifuging for 20 min at 13,000 g at 4°C in order to remove the cell debris. The cleared supernatant was transferred to a fresh tube and total protein was precipitated by the addition of $CHCl_3$/MeOH as previously described (*Gao and Yates, 2019*). Air-dried total protein pellet from each sample was dissolved in 100 mM Tris-HCl pH 8.5, 8 M urea. Protein concentration was measured by BCA protein assay kit (Thermo Fisher). About 100 µg of protein from each sample was digested with trypsin (Promega) at 37°C for 24 hr with an enzyme to protein ratio 1:100 (w/w). After the addition of 96% formic acid to the final concentration of 5%, samples were briefly centrifuged at 13,000 g and peptide-containing supernatant was used for further analysis Peptides were fractionated into eight fractions using High-pH Reversed-Phase Peptide Fractionation Kit (Thermo Fisher). Peptide fractions were dried in SpeedVac (Eppendorf, Germany) and resuspended in 20 µL of 0.1% formic acid.

## Mass spectrometry analysis

Individual fractions of the tryptic peptides were separated by a capillary C18 reverse phase column of the mass-spectrometer-linked Ultimate 3000 HPLC system (Thermo Fisher) using a 90 min gradient (5–85% acetonitrile in water) at 300 nL/min flow. Mass spectrometry data were acquired by an Orbitrap QExactive HF system (Thermo Fisher) with nanospray ESI source using data-dependent acquisition. Raw data were collected and processed by the ProLuCID search engine (*Xu et al., 2015*) and an in-house pipeline.

Raw files were converted into MSn files using in-house converter or mzml files using MSconvert (*Chambers et al., 2012*) and then searched by both ProluCID (*Xu et al., 2015*) and MSFragger (*Kong et al., 2017*) using standard and extended reference proteomes (see below). The identified peptide sequences were first mapped to the in silico-generated extended proteome, then cross-referenced to the standard reference proteome (Uniprot *E. coli* BL21 reference proteome downloaded 2/20/2020). The 'extension' peptides were identified by subtracting all the peptides found in standard reference proteome from the peptides found in the extended proteome. All peptides mapped to the 150 nt-long genomic regions downstream of the stop codons of each annotated ORF and those covering both the tail of annotated ORFs and head of the downstream of the stop codons were reported. All scripts processing the data were written in Python 3.7 and are available at http://git.pepchem.org/gaolab/api_ribo_ext.

The standard reference proteome was constructed using the annotated reference genome of *E. coli*, strain BL21 DE3 (NCBI GenBank: CP001509.3). Extended reference proteome was generated

by amending standard reference proteome with the proteome that could be encoded in three frames in the 150 nt-long genomic regions downstream of the stop codons of the 2100 highly expressed annotated genes (>112 sequence counts or 17.2 RPMs, the median of all genes, in the Ribo-seq dataset) in the Api-treated sample. The bypass of the first downstream stop codon was presumed to take place in 0/ (+1)/(−1) frame using the rules described above for the primary stop codon. The subsequent downstream stop codons were presumed to be translated in 0 frame.

To investigate the readthrough pattern of the primary stop codon, another custom proteome database was constructed with the primary stop codon translated into each of the 20 natural amino acids. The mass spectrometry data were then re-processed the same way as described above except using this new extended database.

To elucidate the changes in the proteome among different conditions, all the proteins identified from the above-mentioned search, regardless of the extension status, were quantified using a semi-quantitative label-free quantitation method using normalized spectral abundance factor (NSAF) to represent the relative expression level of each protein (*Zybailov et al., 2005*). All the low confident proteins, that is, proteins with <5 spectral counts, were removed from the analysis. The rest of the proteins were compared using the NSAF ratio between two conditions and sorted using a gamma distribution cumulative distribution function. The top 50 genes chosen from each comparison were used as input for pathway analysis using String-DB (*Szklarczyk et al., 2019*). A heat map of all the protein expression levels across all conditions was generated using NSAFs (*Figure 5—figure supplement 1A*). Genes and conditions were both clustered using a hierarchical k-mean clustering algorithm (*Gao et al., 2019*). The color represents the relative expression level (NSAFs) of each protein quantified as mentioned above.

## Toeprinting assay

DNA templates for toeprinting containing T7 promoter and an annealing site for the toeprinting primer were generated via PCR by amplifying the corresponding gene sequences from the *E. coli* BL21 genomic DNA. The following DNA primers were used for preparing the respective gene templates: *zntA* (TAATACGACTCACTATAGGGAAACTTAACCGGAGGATGCCATGTCG and GGTTA TAATGAATTTTGCTTATTAACTTTGAACGCAGCAAATTGAGGGGC), *tyrS* (TAATACGACTCACTA TAGGGTTATATACATGGAGATTTTGATGGCAAGC and GGTTATAATGAATTTTGCTTATTAACGAC TGGGCTACCAGCCCCCG), *srmB* (TAATACGACTCACTATAGGGGCCCCACACAGAGGTAGAACA TGACTGTAACG and GGTTATAATGAATTTTGCTTATTAACGGCTTCCAGCAGGCTTTCGTCG), *ybhL* (TAATACGACTCACTATAGGGTATATCTTCAGGAGATCGTCATGGACAGATTCC and GGTTATAA TGAATTTTGCTTATTAACGATTGCAAGCCAGCCCGGGGTTGTACG), *cysS* (TAATACGACTCACTA TAGGGATGTCTAAACGGAATCTTCGATGCTAAAAATCTTC and GGTTATAATGAATTTTGCTTA TTAACTGAATAGGCTTAAATTCCTCTTTTTGGCG). The toeprinting assay was performed as previously described (*Orelle et al., 2013*) using the PURExpress system (New England Biolabs). When needed, Api was added to the final concentration of 50 μM or 2 mM. The final concentration of edeine was 50 μM. Following 10 min of translation, reverse transcription was carried out for 10 min using toeprinting primer NV1 GGTTATAATGAATTTTGCTTATTAAC (*Vazquez-Laslop et al., 2008*). Reverse transcription products were separated on a 6% sequencing gel. The gel was dried, exposed to a phosphorimager screen, and visualized in a Typhoon phosphorimager (GE Healthcare).

## Aggregation assay

*E. coli* BL21Δ*tolC* cells were grown in MOPS media to a density of $A_{600}$ ~0.4 and then treated for 30 min with 2× MIC of either Api, spectinomycin (60 μg/mL), or streptomycin (1 μg/mL). Control antibiotic treatment conditions were those affording maximum protein aggregation (*Ling et al., 2012*). Cells were lysed and protein aggregates were isolated as previously described (*Tomoyasu et al., 2001*). The aggregates were separated with SDS-PAGE and visualized by silver staining.

## Acknowledgements

We thank Dr. Mark Maienschein-Cline (University of Illinois at Chicago) for assistance with the NGS analysis and GALAXY pipeline, Dr. Mo Hu (Northwestern University Feinberg School of Medicine) for help with statistics modeling analysis of the proteomics data, Dr. Cruz-Vera (University of Alabama, Huntsville) for providing anti-RF2 antibodies, and members of Mankin/Vàzquez-Laslop and Polikanov

labs for discussion. This work was supported by the grant from the National Science Foundation (to NV-L and ASM) MCB 1951405, and the grant from the National Institutes of Health R35 GM133416 (to YG); KM was supported in part by Research Training grant 5T32AT007533-07 in Natural Product Complementary and Integrative Health from the Office of the Director, National Institutes of Health, National Center For Complementary and Integrative Health.

# Additional information

## Funding

| Funder | Grant reference number | Author |
| --- | --- | --- |
| National Science Foundation | MCB 1951405 | Alexander S Mankin<br>Nora Vázquez-Laslop |
| National Institutes of Health | R35 GM133416 | Yu Gao |
| National Institutes of Health | 5T32AT007533-07 | Kyle Mangano |

The funders had no role in study design, data collection and interpretation, or the decision to submit the work for publication.

## Author contributions

Kyle Mangano, Formal analysis, Investigation, Methodology, Writing - original draft; Tanja Florin, Conceptualization, Investigation, Methodology, Writing - original draft; Xinhao Shao, Software, Formal analysis, Investigation; Dorota Klepacki, Investigation; Irina Chelysheva, Formal analysis; Zoya Ignatova, Yu Gao, Formal analysis, Supervision, Funding acquisition; Alexander S Mankin, Conceptualization, Supervision, Funding acquisition, Writing - review and editing; Nora Vázquez-Laslop, Conceptualization, Formal analysis, Supervision, Writing - review and editing

## Author ORCIDs

Kyle Mangano (iD) https://orcid.org/0000-0001-9862-2374
Zoya Ignatova (iD) https://orcid.org/0000-0002-9478-8825
Yu Gao (iD) https://orcid.org/0000-0003-3220-4721
Alexander S Mankin (iD) https://orcid.org/0000-0002-3301-827X
Nora Vázquez-Laslop (iD) https://orcid.org/0000-0003-2256-693X

## Decision letter and Author response

Decision letter https://doi.org/10.7554/eLife.62655.sa1
Author response https://doi.org/10.7554/eLife.62655.sa2

# Additional files

## Supplementary files

- Source data 1. Source data from ribosome profiling analysis used in *Figure 2A*.
- Source data 2. SC score correlations.
- Source data 3. Peptides encoded in the downstream mRNA regions found by proteomics.
- Source data 4. Protein abundance in the untreated and Api-exposed cells.
- Source data 5. RNA-seq gene scores.
- Source data 6. Ribo-seq gene scores.
- Source data 7. Api MIC in E coli strains.
- Transparent reporting form

## Data availability

Ribo-seq and RNA-seq data have been deposited in the NCBI Gene Expression Omnibus (GEO) database under accession code GSE150034. Proteomics data can be found in the EMBL-EBI Proteomics Identification database (PRIDE) under accession code PXD019012.

The following datasets were generated:

| Author(s) | Year | Dataset title | Dataset URL | Database and Identifier |
|---|---|---|---|---|
| Mangano K, Florin T, Shao X, Klepacki D, Chelysheva I, Ignatova Z, Gao Y, Mankin AS, Vaź- quez-Laslop N | 2020 | Genome-wide effects of the antimicrobial peptide apidaecin on translation termination | https://www.ncbi.nlm. nih.gov/geo/query/acc. cgi?acc=GSE150034 | NCBI Gene Expression Omnibus, GSE150034 |
| Mangano K, Florin T, Shao X, Klepacki D, Chelysheva I, Ignatova Z, Gao Y, Mankin AS, Vaź- quez-Laslop N | 2020 | Genome-wide effects of the antimicrobial peptide apidaecin on translation termination | https://www.ebi.ac.uk/ pride/archive/projects/ PXD019012 | PRIDE, PXD019012 |

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
