## [Decision Letter]

**Decision letter after peer review:**

[Editors’ note: the authors submitted for reconsideration following the decision after peer review. What follows is the decision letter after the first round of review.]

Thank you for submitting your work entitled "Genome-wide effects of the antimicrobial peptide apidaecin on translation termination" for consideration by *eLife*. Your article has been reviewed by two peer reviewers, including Rachel Green as the Reviewing Editor and Reviewer #1, and the evaluation has been overseen by a Reviewing Editor and a Senior Editor.

We now have two reviews on your manuscript, one from myself, and the other from an expert in the field. While we normally would obtain three reviews, the congruence of these reviews was sufficient to return with a response. Overall, both reviewers appreciated the contribution that this work has the potential to make in defining the specificity and impact of apidaecin (Api) on bacterial ribosomes in living cells. Previous biochemical work from the group showed that Api traps deacylated tRNA in the P site, post peptide release, but where RFs remain bound in the A site. This work also showed that there was a broad shut down of termination in the lysates which was interpreted to reflect the sequestration of the RFs by the Api-bound ribosomes. This work uses ribosome profiling and mass spec to demonstrate that ribosomes do indeed accumulate (and stack) at stop codons in cells treated with Api, and that they occasionally read through the stop codon and generate new peptides (presumably those ribosomes that are stalled prior to peptide release). Overall, the work is strong but lacks much in the way of new insight into the role of Api in stopping translation. Both reviewers agreed that there are several points that would need to be addressed in order for this study to make novel contributions to our understanding. Given that addressing these points and the reviews below will require additional experimentation and is expected to take more than two months, the manuscript is being rejected at this time. However, please consider re-submitting to *eLife* if you are able to address the comments.

1) The analysis of specificity in Figure 3 seemed overstated. First, both reviewers were somewhat concerned that the SC metric included 3' UTR sequence in the analysis, given that readthrough on different mRNAs could be variable, and the location of next stop codon would also be variable, and that these features could differentially impact the SC values – also not sure that it makes any sense to include. A second concern was whether the differences in variance identified in panel B were real or rather were exaggerated by presentation of these data in linear rather than log format. Finally, it is critical that the motifs that appear to arise in the Api-treated cells don't show up in the untreated cells – otherwise it has nothing to do with Api and more to do with natural readthrough propensity transcriptome wide.

2) The collection of ribosomes at stop codons is likely heterogeneous – some are stalled with Api and an RF bound, the others contain peptidyl-tRNA and no RF. Yet these studies fail to separate these two classes and to analyze differences. Preparing ribosome profiling libraries with a salt wash would yield libraries enriched in ribosomes containing peptidyl-tRNAs. Alternatively, overexpression of RF1 might clear a subset of ribosomes and establish that the model is correct (not enough free RF1 to function in termination). The Pm experiment does not make sense. Why do ribosomes in the queue and the 3' UTR clear, but not at the stop codons. Those occupied by RFs might be resistant to Pm but those trapped by a lack of RF should be sensitive. These data make me concerned that the model is not correct.

3) Figuring out whether the overall model is correct should be possible using simple polysome analysis. The prediction is that RFs should be enriched in polysomal fractions when cells are treated with Api. This simple experiment could lend strong support to the models derived from in vitro experiments.

Reviewer #1:

This manuscript using ribosome profiling to characterize the impact of the antimicrobial peptide apidaecin (Api) on bacterial ribosomes in living cells. Previous structural studies have shown that Api binds in the vacant nascent peptide exit tunnel of the ribosome, approaching the PTC, but not directly occupying the catalytic site. Biochemical experiments in vitro showed that Api traps deacylated tRNA in the P site, post peptide release, but where the RFs remain bound in the A site. As such the ribosome is trapped in a post-release state, where the release of the nascent peptide allows binding of Api, thus trapping RF1/2 on the ribosome. However, it was also clear that treatment of lysates with Api led to the accumulation of peptidyl-RNAs, as though the actual release reaction were also inhibited. These ideas led to the model that trapping of RF1/2 on ribosomes would lead to its depletion in the cell and genome-wide build up of ribosomes at stop codons with no way to properly terminate. Here these models are tested.

The data presented are straightforward. The authors present ribosome profiling data with Api, with and without puromycin to independently release nascent peptides, and look at ribosome accumulation throughout the transcriptome. As anticipated, ribosomes accumulate at stop codons, and exhibit queuing behind the lead ribosome with a periodicity of about 27 nts. The authors analyze the stop codon (SC) peak size, observing broad increases for most genes, and then attempted to define some features that correlated with especially increased SC scores (including a Glycine residue at the terminal amino acid of the peptide chain, for example). The authors then characterized stop codon readthrough (RT) using the same data and the same types of analyses. Evidence for RT was corroborated by shotgun mass spec data revealing peptides that represent sequences found in the 3' UTRs. These were disproportionately in frame, though not dramatically so (and there was no statistical analysis to support this). These ideas were consistent with their initial model that depletion of RFs by Api stalled ribosomes would result in broad inhibition of termination, though those ribosomes would not be blocked per se by Api, and if they can escape the stop codon can continue elongating. The final section of the manuscript focuses on the regulation of various bacterial rescue pathways under treatment with Api, but arguing ultimately that none of these pathways can help deal with Api since all of them depend in some way on a termination reaction that utilizes a release factor or related protein (arfB), and this factor would similarly be inhibited by Api.

Overall, the manuscript was clearly written and presented some clear data on the mechanism of action of Api in vivo that is appealingly consistent with the models presented. However, the manuscript did not fully explore features of terminating ribosomes that might have better informed their models. The authors used Pm to show that elongating ribosomes were released from the ORF and to some extent from the 3'UTR (though many are left behind), though ribosomes at stop codons are resistant. These data are consistent with the idea that some ribosomes at the stop codon are bound by RF1/2 and are thus resistant to ribosome dissociation by Pm. The authors also argue however that there are different types of ribosomes stuck at stop codons.… those trapped with Api and those trapped because RF1/2 has been depleted. These should be differently sensitive to salt washing, as one class carries nascent peptide while the other does not. This experiment seems important since the analysis of the SC score will be confounded by these two populations of ribosomes, and so it is not clear whether the Glycine residue contributes to Api action on the stop codon or to preferential sensitivity to RF depletion. Another weakness of the manuscript is in the absence of any orthogonal approaches to define the model that they favor (depletion of RF1/2 on Api bound ribosomes). If RF1/2 are sequestered on ribosomes in the presence of Api, it seems likely that analysis of the distribution of RF1/2 on a polysome profile could provide strong support for such a model that would strengthen the conclusions of the manuscript.

The SC metric was defined by the ribosome density within the last three codons of the gene relative to average ORF density including the 40 nts of the 3 UTR downstream of the stop codon. This calculation was not clear to me – why do the authors include 3' UTR in this analysis.

Reviewer #2:

This manuscript by Mangano et al. describes ribosome profiling experiments that reveal the in vivo activities of apidaecin (Api), an antibiotic peptide in the PrAMP family. They show that treating *E. coli* cells with Api leads to dramatic enrichment of ribosome density at stop codons as well as the formation of queues of stalled ribosomes upstream of stop codons. Using ribosome profiling and proteomic approaches, they show that ribosomes often read through stop codons or frameshift to continue elongation. These data are largely confirmatory, supporting the model of Florian et al., 2017 that Api binds in the ribosome exit tunnel and traps release factors on the ribosome, titrating them away so that translational termination is globally inhibited.

I have a few concerns about the analyses as described below. In particular, they should test whether effects seen on UGA and other nearby sequence features are specific to Api or are more generally true during translational termination.

Subsection “Api acts as a global inhibitor of translation termination”: The authors state the increase in the stop codon peak shows that Api is "a potent global inhibitor of translation termination." It is possible that the peak at stop codons comes from pre-release ribosomes or post-release ribosomes that have not yet been recycled. In yeast, stop codon peaks arise primarily from post-release ribosomes (meaning recycling is slow, see Schuller and Green, 2017). The authors should state both of these options here and acknowledge that profiling cannot distinguish between them. Then they can argue later based on the readthrough and frameshifting data that these are in fact pre-release ribosomes.

Subsection “Api acts as a global inhibitor of translation termination”: The authors write "the Api treated cells show a much broader distribution compared to those of the untreated control." This is true on an absolute scale, of course, as we see in Figure 3B, but on a log_2_ scale (Figure 3A) the distributions seem quite similar. Api increases the signal, to be sure, but the relative distribution surrounding the signal remains the same. I argue that the relative distribution is what they really want, since it gives information about how local mRNA or peptide context affects termination.

Figure 3B and subsection “Api acts as a global inhibitor of translation termination”: The Api treated cells have fewer stalled ribosomes at UGA stop codons than UAA or UAG and the authors argue that Api might trap RF2 less efficiently. But this assumes that this effect is specific to Api. It looks to me like in Figure 3B that the same is true of untreated cells (less stalling at UGA than the others), arguing it is not an Api effect, but something intrinsic about UGA and RF2. This would be clearer if Figure 3B had a log_2_ scale on the y-axis.

Figure 3C and D show sequence features associated with high levels of stalling at stop codons in Api treated cells. What about untreated cells? This analysis would reveal if these features are general to termination or if they are an effect of Api specifically.

Subsection “Api acts as a global inhibitor of translation termination”: The authors cite Oh, 2011 to show that bacterial ribosome footprints are ~27 nt, but this depends on how they are prepared. What is the major footprint size in their data?

Subsection “Api-induced translation arrest at stop codons results in queuing of the elongating Ribosomes”: I don't understand the Pm results. They argue that Pm removes elongating ribosomes in the queue but leaves the stop codon peak intact. Indeed, the data in Figure 2—figure supplement 1A support this. But their model is that the ribosomes at stop codons still have peptidyl-tRNA (i.e. pre-release). So why are they not released by Pm?

Figure 4: Can the authors use the profiling data to identify mRNA features associated with read-through or frameshifting in the treated and untreated samples? Presumably the features are the same in both samples, just amplified by Api?

Figure 5C: is this really showing ribosomes at the end of transcripts? I'm not convinced. They could use publicly available Term-seq data (Sorek, 2016) or operon annotations to determine the 3'-ends. And a metagene plot would be more convincing than one or two anecdotal examples.

Subsection “Api treatment distorts the cell proteome and activates cellular ribosome rescue Systems”: In general start codon peaks vary in intensity, it's hard to say that this is really due to the antibiotic. One of the toeprints looks good, the other weak, but 2 mM is quite high.

Discussion section: But enhanced termination with -3 Arg should give you a lower SC score, which is backwards. -3 Arg and -1 Gly play a role in elongation stalling in the GIRAG sequence in SecM.

---

## [Author Response]

[Editors’ note: the authors resubmitted a revised version of the paper for consideration. What follows is the authors’ response to the first round of review.]

We now have two reviews on your manuscript, one from myself, and the other from an expert in the field. While we normally would obtain three reviews, the congruence of these reviews was sufficient to return with a response. Overall, both reviewers appreciated the contribution that this work has the potential to make in defining the specificity and impact of apidaecin (Api) on bacterial ribosomes in living cells. Previous biochemical work from the group showed that Api traps deacylated tRNA in the P site, post peptide release, but where RFs remain bound in the A site. This work also showed that there was a broad shut down of termination in the lysates which was interpreted to reflect the sequestration of the RFs by the Api-bound ribosomes. This work uses ribosome profiling and mass spec to demonstrate that ribosomes do indeed accumulate (and stack) at stop codons in cells treated with Api, and that they occasionally read through the stop codon and generate new peptides (presumably those ribosomes that are stalled prior to peptide release). Overall, the work is strong but lacks much in the way of new insight into the role of Api in stopping translation. Both reviewers agreed that there are several points that would need to be addressed in order for this study to make novel contributions to our understanding. Given that addressing these points and the reviews below will require additional experimentation and is expected to take more than two months, the manuscript is being rejected at this time. However, please consider re-submitting to eLife if you are able to address the comments.

Thank you for carefully examining our work. Following your recommendation, we are resubmitting to *eLife* the expanded and revised manuscript.

1) The analysis of specificity in Figure 3 seemed overstated. First, both reviewers were somewhat concerned that the SC metric included 3' UTR sequence in the analysis, given that readthrough on different mRNAs could be variable, and the location of next stop codon would also be variable, and that these features could differentially impact the SC values – also not sure that it makes any sense to include. A second concern was whether the differences in variance identified in panel B were real or rather were exaggerated by presentation of these data in linear rather than log format. Finally, it is critical that the motifs that appear to arise in the Api-treated cells don't show up in the untreated cells – otherwise it has nothing to do with Api and more to do with natural readthrough propensity transcriptome wide.

We agree that our original SC metric was confusing and, following the reviewers’ suggestions, we have changed the scoring metrics. Specifically, we have excluded the 3’UTR reads calculating the new SC score by dividing the ribosome density (number of reads) at the last three codons by the total density in the entire ORF – from start to stop. Using the new metrics, the stop codon identity shows a milder effect upon the SC score and, therefore, we have deemphasized this point. Furthermore, to avoid the confusion noted by the reviewers, the data are now presented as a log_2_ transformed violin plot and are shown not in the main text figure, but in Figure 3—figure supplement 2.

In addition, following the reviewers advise, we now include the pLogo analysis of the C-terminal amino acid sequences of the high-SC scoring genes not only for the Api sample, but also for the untreated control, all based on the new SC scoring metrics. In this case, the statistically significant difference still remains. The results are included in panel B of the revised Figure 3.

2) The collection of ribosomes at stop codons is likely heterogeneous – some are stalled with Api and an RF bound, the others contain peptidyl-tRNA and no RF. Yet these studies fail to separate these two classes and to analyze differences. Preparing ribosome profiling libraries with a salt wash would yield libraries enriched in ribosomes containing peptidyl-tRNAs. Alternatively, overexpression of RF1 might clear a subset of ribosomes and establish that the model is correct (not enough free RF1 to function in termination). The Pm experiment does not make sense. Why do ribosomes in the queue and the 3' UTR clear, but not at the stop codons. Those occupied by RFs might be resistant to Pm but those trapped by a lack of RF should be sensitive. These data make me concerned that the model is not correct.

We fully agree that it would be interesting to know the ratio of pre- and post-hydrolysis ribosomes at stop codons in the Api-treated cells. However, this is a difficult question to address experimentally. Although ribosome profiling with a high-salt wash is an interesting idea, it is unknown whether Api bound ribosomes, where tunnel-lodged Api interacts with the P-site tRNA and the A-site RF1/2 would be susceptible to salt-induced dissociation. Furthermore, because the approaches for computing stop codon occupancy (SC score or metagene) use relative, not absolute values, any changes resulting from salt wash could be difficult to interpret.

The other suggestion – to overexpress RF1 – is also interesting. However, for the aforementioned reasons it is also unlikely to clarify the ratio of the pre- and post release ribosome complexes at stop codons. Furthermore, overexpression of class 1 release factors is known to render cells more resistant to Api (Matsumoto et al., 2017); the MIC difference would make it difficult to find the conditions for an accurate comparison of Api effects upon wt and RF-overexpressing cells.

Being unable to address the ratio of pre-and post-release ribosomes experimentally, we instead calculated the fraction of the stop codon footprints relative to all the mapped footprints and found that ~11% of them map to stop codons in the Api-treated cells (as opposed to ~2% in the control). The resulting fraction of stop codon-associated ribosomes roughly equals the estimated number of cellular RFs (RF2, to be more precise), in exponentially growing cells. Therefore, we believe that most of the ribosomes found at stop codons are associated with RFs and thus, have released the nascent protein.

Accordingly, we have rectified our model by proposing that the majority of stop codon ribosomes are post-release and Api-bound. While this has not changed the implications of our model, it has clarified it and we are thankful for bringing these considerations to our attention. We added a new figure (Figure 6) to illustrate our model.

We believe that out metagene plot of the Pmn-treated Api sample does make perfect sense. We appreciate, however, that some of the confusion could stem from the apparent increase in the magnitude of the ribosome peak at stop codons upon Pmn treatment (former Figure 2—figure supplement 1). Because the drug is expected to remove the ribosomes stalled in a pre-release state, it is natural to expect that the stop codon peak should decrease when the lysates are treated with Pmn. However, our metagene metrics utilizes the *relative* occupancy, where the scores are computed as a fraction of ribosomes at each nucleotide within the analyzed window. Because Pmn removes a significant fraction of queued and 3’-UTR ribosomes, the fraction of ribosomes at stop codon increases even if some of them are removed by Pmn.To avoid such a confusion, in the revised figure (Figure 2—figure supplement 2) we use a cartoon to illustrate the ribosomes that are removed by Pmn and use the % of normalized RPM for the y axis. We further clarify this point in the text and figure legend.

3) Figuring out whether the overall model is correct should be possible using simple polysome analysis. The prediction is that RFs should be enriched in polysomal fractions when cells are treated with Api. This simple experiment could lend strong support to the models derived from in vitro experiments.

This is a great idea, although we have implemented it in a slightly different format. Instead of analyzing RFs in the polysome fractions, which could be difficult to interpret, we have instead tested the amount of free RF (RF2, to be precise) in the cytoplasm of the Api-treated cells. After lysing the cells and pelleting ribosomes through a sucrose cushion, we tested for the amount of free RF2 in the supernatant. While a robust RF2 band is evident in the supernatant of untreated control cells, free RF2 was essentially undetectable in the Api-treated cells.This result, which is presented in Figure 4—figure supplement 3, strongly confirms our model that Api treatment leads to depletion of free class 1 RFs.

Reviewer #1:This manuscript using ribosome profiling to characterize the impact of the antimicrobial peptide apidaecin (Api) on bacterial ribosomes in living cells. Previous structural studies have shown that Api binds in the vacant nascent peptide exit tunnel of the ribosome, approaching the PTC, but not directly occupying the catalytic site. Biochemical experiments in vitro showed that Api traps deacylated tRNA in the P site, post peptide release, but where the RFs remain bound in the A site. As such the ribosome is trapped in a post-release state, where the release of the nascent peptide allows binding of Api, thus trapping RF1/2 on the ribosome. However, it was also clear that treatment of lysates with Api led to the accumulation of peptidyl-RNAs, as though the actual release reaction were also inhibited. These ideas led to the model that trapping of RF1/2 on ribosomes would lead to its depletion in the cell and genome-wide build up of ribosomes at stop codons with no way to properly terminate. Here these models are tested.The data presented are straightforward. The authors present ribosome profiling data with Api, with and without puromycin to independently release nascent peptides, and look at ribosome accumulation throughout the transcriptome. As anticipated, ribosomes accumulate at stop codons, and exhibit queuing behind the lead ribosome with a periodicity of about 27 nts. The authors analyze the stop codon (SC) peak size, observing broad increases for most genes, and then attempted to define some features that correlated with especially increased SC scores (including a Glycine residue at the terminal amino acid of the peptide chain, for example). The authors then characterized stop codon readthrough (RT) using the same data and the same types of analyses. Evidence for RT was corroborated by shotgun mass spec data revealing peptides that represent sequences found in the 3' UTRs. These were disproportionately in frame, though not dramatically so (and there was no statistical analysis to support this). These ideas were consistent with their initial model that depletion of RFs by Api stalled ribosomes would result in broad inhibition of termination, though those ribosomes would not be blocked per se by Api, and if they can escape the stop codon can continue elongating. The final section of the manuscript focuses on the regulation of various bacterial rescue pathways under treatment with Api, but arguing ultimately that none of these pathways can help deal with Api since all of them depend in some way on a termination reaction that utilizes a release factor or related protein (arfB), and this factor would similarly be inhibited by Api.Overall, the manuscript was clearly written and presented some clear data on the mechanism of action of Api in vivo that is appealingly consistent with the models presented. However, the manuscript did not fully explore features of terminating ribosomes that might have better informed their models. The authors used Pm to show that elongating ribosomes were released from the ORF and to some extent from the 3'UTR (though many are left behind), though ribosomes at stop codons are resistant. These data are consistent with the idea that some ribosomes at the stop codon are bound by RF1/2 and are thus resistant to ribosome dissociation by Pm. The authors also argue however that there are different types of ribosomes stuck at stop codons.… those trapped with Api and those trapped because RF1/2 has been depleted. These should be differently sensitive to salt washing, as one class carries nascent peptide while the other does not. This experiment seems important since the analysis of the SC score will be confounded by these two populations of ribosomes, and so it is not clear whether the Glycine residue contributes to Api action on the stop codon or to preferential sensitivity to RF depletion.

We believe we have addressed most of these concerns in our comments above. As we indicated, we do not have a convincing way to assess the fraction of pre-and post-release ribosomes stalled at stop codons in the Api-treated cells, but our estimates based on the profiling data argue that the majority of them are stalled in the post-release Api-bound state. Nevertheless, we are not sure we have sufficient data to conclude whether C-terminal glycine increases the fraction of pre-or post-release ribosomes and we openly state this in the revised manuscript (subsection “Api treatment leads to pervasive stop codon readthrough”).

Another weakness of the manuscript is in the absence of any orthogonal approaches to define the model that they favor (depletion of RF1/2 on Api bound ribosomes). If RF1/2 are sequestered on ribosomes in the presence of Api, it seems likely that analysis of the distribution of RF1/2 on a polysome profile could provide strong support for such a model that would strengthen the conclusions of the manuscript.The SC metric was defined by the ribosome density within the last three codons of the gene relative to average ORF density including the 40 nts of the 3 UTR downstream of the stop codon. This calculation was not clear to me – why do the authors include 3' UTR in this analysis.

We think that distribution of RF1/2 on polysome profiles would be hard to interpret.

However, an aforementioned conceptually-similar experiment inspired by this comment (Figure 4—figure supplement 3) convincingly shows sequestration of RFs on the ribosomes in Api-treated cells. In addition, as indicated above, we have changed the SC metric, now excluding the 3’ UTR.

Reviewer #2:This manuscript by Mangano et al. describes ribosome profiling experiments that reveal the in vivo activities of apidaecin (Api), an antibiotic peptide in the PrAMP family. They show that treating *E. coli* cells with Api leads to dramatic enrichment of ribosome density at stop codons as well as the formation of queues of stalled ribosomes upstream of stop codons. Using ribosome profiling and proteomic approaches, they show that ribosomes often read through stop codons or frameshift to continue elongation. These data are largely confirmatory, supporting the model of Florian et al., 2017 that Api binds in the ribosome exit tunnel and traps release factors on the ribosome, titrating them away so that translational termination is globally inhibited.

We thank the reviewer for carefully evaluating our manuscript. Nevertheless, we respectfully disagree that our data are largely confirmatory. Firstly, prior to this work, most of what we knew about Api action came from in vitro biochemical experiments or microbiological MIC testing and the proposed models required large leaps of faith. The previous in vitro experiments, which involved artificial substrates, unnatural settings, and non-translating ribosomes were incapable to reveal or even predict the range and magnitude of the in vivo effects. Indeed, neither the pronounced ribosome queuing, nor the massive stop codon readthrough and synthesis of the extended proteins could be revealed in vitro. As an illustration, on the basis of the published biochemical data, we and others thought that Api could be directly used for mapping translation termination sites genome-wide. Little did we know! It is only with our new data that we could think of possible approaches, e.g. Pmn treatment, that could be exploited for achieving such a goal. Similarly, translation all the way to the mRNA 3’ ends and the massive, but futile, engagement of the rescue systems were practically impossible to predict. Thus, we think that our study significantly advances the understanding of how a specific inhibitor of translation termination affects global cellular translation.

I have a few concerns about the analyses as described below. In particular, they should test whether effects seen on UGA and other nearby sequence features are specific to Api or are more generally true during translational termination.

We thank the reviewer for bringing up this point. Indeed, after implementing the new scoring metric, we realized that firstly, the UGA-specific effects are less strong than we originally thought and secondly, they generally match the trend observed in the untreated cells. Therefore, in the revised manuscript we have significantly deemphasized the stalling effect at the UGA codons. On the other hand, the influence of C-terminal glycine appears to be authentic and Api-specific (revised Figure 3B). The comparison with the untreated control, which following the reviewer’s suggestion we have introduced in the revised manuscript, helps to make this point stronger.

Subsection “Api acts as a global inhibitor of translation termination”: The authors state the increase in the stop codon peak shows that Api is "a potent global inhibitor of translation termination." It is possible that the peak at stop codons comes from pre-release ribosomes or post-release ribosomes that have not yet been recycled. In yeast, stop codon peaks arise primarily from post-release ribosomes (meaning recycling is slow, see Schuller and Green, 2017). The authors should state both of these options here and acknowledge that profiling cannot distinguish between them. Then they can argue later based on the readthrough and frameshifting data that these are in fact pre-release ribosomes.

Because the fraction of stop-codon bound ribosomes roughly matches the number of

RFs, we believe that most of the stop codon footprints are accounted for by the post-release, Api- and RF-bound ribosomes. However, we agree that as in the untreated cells, some amount of the stop-codon associated ribosomes are there because of the slow recycling. We mentioned this possibility in the revised manuscript (subsection “Api treatment leads to pervasive stop codon readthrough”) and added the recommended reference.

Subsection “Api acts as a global inhibitor of translation termination”: The authors write "the Api-treated cells show a much broader distribution compared to those of the untreated control." This is true on an absolute scale, of course, as we see in Figure 3B, but on a log_2_ scale (Figure 3A) the distributions seem quite similar. Api increases the signal, to be sure, but the relative distribution surrounding the signal remains the same. I argue that the relative distribution is what they really want, since it gives information about how local mRNA or peptide context affects termination.

This is a very good point. We got caught up thinking about the SC and RT scores in non-logged scales. After log_2_ transformation, the SC score distributions are indeed similar and in the revised manuscript we have deemphasized the difference in the range of distribution between the Api and control samples. Importantly, some of the context-specificity trends appear to be Api-specific indicating that Api does not simply amplify the intrinsic difference in the termination rate at stop codons of different genes.

Figure 3B and subsection “Api acts as a global inhibitor of translation termination”: The Api treated cells have fewer stalled ribosomes at UGA stop codons than UAA or UAG and the authors argue that Api might trap RF2 less efficiently. But this assumes that this effect is specific to Api. It looks to me like in Figure 3B that the same is true of untreated cells (less stalling at UGA than the others), arguing it is not an Api effect, but something intrinsic about UGA and RF2. This would be clearer if Figure 3B had a log_2_ scale on the y-axis.

As suggested by the reviewer, we have converted the scale to log_2_ and have deemphasized the stop codon-specificity of the observed effects.

Figure 3C and D show sequence features associated with high levels of stalling at stop codons in Api treated cells. What about untreated cells? This analysis would reveal if these features are general to termination or if they are an effect of Api specifically.

We completely agree with the reviewer that comparison with the untreated control is critical and have included it in all the relevant figures, including Figure 3.

Subsection “Api acts as a global inhibitor of translation termination”: The authors cite Oh, 2011 to show that bacterial ribosome footprints are ~27 nt, but this depends on how they are prepared. What is the major footprint size in their data?

The read length distribution in our profiling experiments was fairly broad: it peaked at 27 nt but had a long tail of longer reads. In retrospect, and in view of more recent papers, analyzing longer reads from the di- and trisome peaks could be informative and we are considering carrying such experiment(s) in future. Meanwhile, in the revised manuscript we have included a supplementary figure showing read length distribution (Figure 2—figure supplement 1).

Subsection “Api-induced translation arrest at stop codons results in queuing of the elongating Ribosomes”: I don't understand the Pm results. They argue that Pm removes elongating ribosomes in the queue but leaves the stop codon peak intact. Indeed the data in Figure 2—figure supplement 1A support this. But their model is that the ribosomes at stop codons still have peptidyl-tRNA (i.e. pre-release). So why are they not released by Pm?

We assume full responsibility for not explaining this result in sufficiently clear terms in the original manuscript, but have done our best to fix this problem in the revised paper. Please see our response to a similar comment of the editor (above). In brief, Pmn treatment likely does remove the pre-release fraction of stop codon-associated ribosomes. The confusing increase in the height of the post-Pmn peak at stop codons is due to the fact that it represents *normalized*, not absolute, reads. Removal of the queued and 3’-UTR ribosomes leads to an increase in the *relative* amount of the stop codon-bound ribosomes. We have clarified this point by adding a cartoon to the figure (Figure 2—figure supplement 2), using % of normalized reads for the y axis and providing a better explanation in the legend and the manuscript text.

Figure 4: Can the authors use the profiling data to identify mRNA features associated with read-through or frameshifting in the treated and untreated samples? Presumably the features are the same in both samples, just amplified by Api?

This is a good point. However, after re-examining the data and comparing the Api samples with the untreated controls we were unable to identify any strong trends that would be amplified or altered by Api. Therefore, we avoided discussing them in the paper.

Figure 5C: is this really showing ribosomes at the end of transcripts? I'm not convinced. They could use publicly available Term-seq data (Sorek, 2016) or operon annotations to determine the 3'-ends. And a metagene plot would be more convincing than one or two anecdotal examples.

We thank the reviewer for this insightful comment and suggestion of using Term-seq data. The reanalysis of our results using the *E. coli* Term-seq data showed a clear and robust increase in the occupancy of the mRNA 3’-proximal segments by the translating ribosomes. We included these new results in panel D of the main Figure 5.

Subsection “Api treatment distorts the cell proteome and activates cellular ribosome rescue Systems”: In general start codon peaks vary in intensity, it's hard to say that this is really due to the antibiotic. One of the toeprints looks good, the other weak, but 2 mM is quite high.

The reviewer is correct that start codon peaks vary significantly, both in the control and in Api samples. It is also true that we could reproduce in vitro some of the start codons effects only at a very high concentration of Api (2 mM). This is the reason why we discuss the start codon effects only tentatively. Nevertheless, we felt it was important to bring to the reader’s attention a possibility that due to its binding in the vacant exit tunnel, Api could have other effects on translation rather than only RF trapping.

Discussion section: But enhanced termination with -3 Arg should give you a lower SC score, which is backwards. -3 Arg and -1 Gly play a role in elongation stalling in the GIRAG sequence in SecM.

We agree that the original explanation was confusing. We have now streamlined the discussion of the context-specific effects.